# Neuroanatomy of a hydrothermal vent shrimp provides insights into the evolution of crustacean integrative brain centers

Julia Machon[1]*, Jakob Krieger[2], Rebecca Meth[2], Magali Zbinden[1], Juliette Ravaux[1], Nicolas Montagné[3], Thomas Chertemps[3], Steffen Harzsch[2]*

[1]Sorbonne Université, UMR CNRS MNHN 7208 Biologie des organismes et écosystèmes aquatiques (BOREA), Equipe Adaptation aux Milieux Extrêmes, Paris, France; [2]Department of Cytology and Evolutionary Biology, University of Greifswald, Zoological Institute and Museum, Greifswald, Germany; [3]Sorbonne Université, UPEC, Univ Paris Diderot, CNRS, INRA, IRD, Institute of Ecology & Environmental Sciences of Paris (iEES-Paris), Paris, France

**Abstract** Alvinocaridid shrimps are emblematic representatives of the deep hydrothermal vent fauna at the Mid-Atlantic Ridge. They are adapted to a mostly aphotic habitat with extreme physicochemical conditions in the vicinity of the hydrothermal fluid emissions. Here, we investigated the brain architecture of the vent shrimp *Rimicaris exoculata* to understand possible adaptations of its nervous system to the hydrothermal sensory landscape. Its brain is modified from the crustacean brain ground pattern by featuring relatively small visual and olfactory neuropils that contrast with well-developed higher integrative centers, the hemiellipsoid bodies. We propose that these structures in vent shrimps may fulfill functions in addition to higher order sensory processing and suggest a role in place memory. Our study promotes vent shrimps as fascinating models to gain insights into sensory adaptations to peculiar environmental conditions, and the evolutionary transformation of specific brain areas in Crustacea.
DOI: https://doi.org/10.7554/eLife.47550.001

*For correspondence:
julia.machon@live.fr (JM);
steffen.harzsch@uni-greifswald.de (SH)

**Competing interests:** The authors declare that no competing interests exist.

## Introduction

The alvinocaridid shrimps were discovered in 1985 during a mission of the deep submersible vehicle ALVIN (*Rona et al., 1986*) and are now known to be widely distributed representatives of the deep hydrothermal vent fauna along the Mid-Atlantic Ridge (MAR; *Desbruyères et al., 2001*; *Desbruyères et al., 2000*; *Gebruk et al., 1997*; *Segonzac et al., 1993*). Active vents are dynamic environments, where geothermally heated seawater, the hydrothermal fluid, discharges from chimneys and cracks in the seafloor. At the MAR, vents occur from 850 to 4080 m depth and the pure hydrothermal fluid, which may be up to 350°C, is anoxic, acid, and enriched in potentially toxic minerals and dissolved gases (*Charlou et al., 2010*; *Charlou et al., 2002*; *Charlou et al., 2000*). Hydrothermal vent habitats, in addition to high hydrostatic pressure and the complete absence of sunlight, are characterized by steep gradients of temperature and concentration of chemicals (*Bates et al., 2010*; *Johnson et al., 1988*; *Johnson et al., 1986*; *Le Bris et al., 2005*). Vent organisms are well adapted to these physicochemical conditions, and alvinocaridid shrimps colonize in high abundance the walls of active chimneys, where the hydrothermal fluid mixes with the surrounding cold (4°C) and oxygenated seawater. Vent ecosystems rely on chemoautotrophic bacteria as primary producers, which convert reduced chemicals through oxidation, thus providing the energy to fix carbon and to

**eLife digest** Oceanic vents are areas where hot gases and liquids emerge from cracks and chimneys on the seafloor. These fluids can be as hot as 350°C and are rich in potentially toxic chemicals. Nevertheless, they are the key energy source of many animals that make the vents their home. Vents can be found thousands of meters under sea level, where no sunlight penetrates, so the animals living there must use senses other than vision. As an example, the vent shrimp *Rimicaris exoculata*, which is used as a vent model animal, was thought to orient itself by sensing chemicals in the vents through their sense of smell.

Machon et al. investigate whether vent shrimps possess particular abilities to detect the chemical landscape of the hydrothermal environment, and describe the brain structure and associated sensory systems of *R. exoculata*. Since the brain in these shrimps is subdivided into regions devoted to different functions, if one of their senses were used more than the others the region devoted to this sense should be bigger or structurally different.

When the anatomy of the brain centers in *R. exoculata* was compared to that of its shallow-water relatives, there was no suggestion that the vent shrimps had an advanced ability to sense chemicals. Rather, a striking feature of the brain of the vent shrimps is the volume and structure of their higher brain centers, which integrate all of their sensory information. It is possible that these regions are also involved in other brain functions as well, since they take up an especially high proportion of the brain. Machon et al. found similarities between *R. exoculata* and other crustaceans that have sophisticated navigation skills so they hypothesize that integrative brain centers in vent shrimps could play a role in place memory.

The findings provide new insights for biologists studying animals associated with deep hydrothermal vents and are also important for neuroscientists interested in brain function and evolution. Future studies should focus on senses of the vent shrimp other than smell to ultimately understand the lifestyle and long-term survival of vent animals.

DOI: https://doi.org/10.7554/eLife.47550.002

produce organic matter that serves as a nutritional basis for primary consumers (*Fisher et al., 2007*; *Jannasch and Mottl, 1985*; *Ponsard et al., 2013*; *Van Dover, 2000*).

The shrimp *Rimicaris exoculata* (*Williams and Rona, 1986*) is the most intensely studied vent crustacean due to its high abundance at most sites along the MAR and its singular lifestyle (*Figure 1A,B*; *Desbruyères et al., 2001*; *Gebruk et al., 1997*; *Segonzac et al., 1993*; *Van Dover et al., 1988*). Specimens of *R. exoculata* are found from 1600 to 4000 m depth (*Lunina and Vereshchaka, 2014*) and they form massive aggregations in the vicinity of the chimneys, with up to 3000 ind.m$^{-2}$ (*Segonzac et al., 1993*). This species is a strict primary consumer, relying on ectosymbiotic bacteria harbored in its enlarged branchial chambers, through a direct nutritional transfer of bacterial carbon products by trans-tegumental absorption (*Corbari et al., 2008*; *Petersen et al., 2010*; *Ponsard et al., 2013*; *Zbinden et al., 2004*). The associated bacterial metabolic activities include oxidation of sulfide, iron, methane and hydrogen, suggesting that *R. exoculata* symbionts could have both nutritional and detoxifying roles for the shrimp (*Hügler et al., 2011*; *Zbinden et al., 2008*). Hence, this species is strictly dependent on hydrothermal fluid emissions to supplement its symbionts with reduced compounds, and might possess specific sensory abilities for this purpose. Because *R. exoculata* preferentially lives close to the hydrothermal fluids, the shrimp constantly has to cope with steep temperature gradients ranging approximately from 4°C to 40°C (*Cathalot and Rouxel, 2018*), and its sensory system might be tuned to efficiently probe this dynamic thermal environment.

A fundamental question regarding vent shrimp's environment and lifestyle is how they detect hydrothermal emissions and further select their microhabitat. Both abiotic and biotic factors are important to determine the animal's local distribution at hydrothermal vent sites (*Le Bris et al., 2005*; *Luther et al., 2001*). Several studies showed that *R. exoculata* possesses a range of morphological, anatomical and physiological adaptations to the hydrothermal environment, related for instance to ectosymbiosis with bacteria (*Casanova et al., 1993*; *Ponsard et al., 2013*; *Zbinden et al., 2004*), respiration in hypoxic conditions (*Hourdez and Lallier, 2006*; *Lallier and*

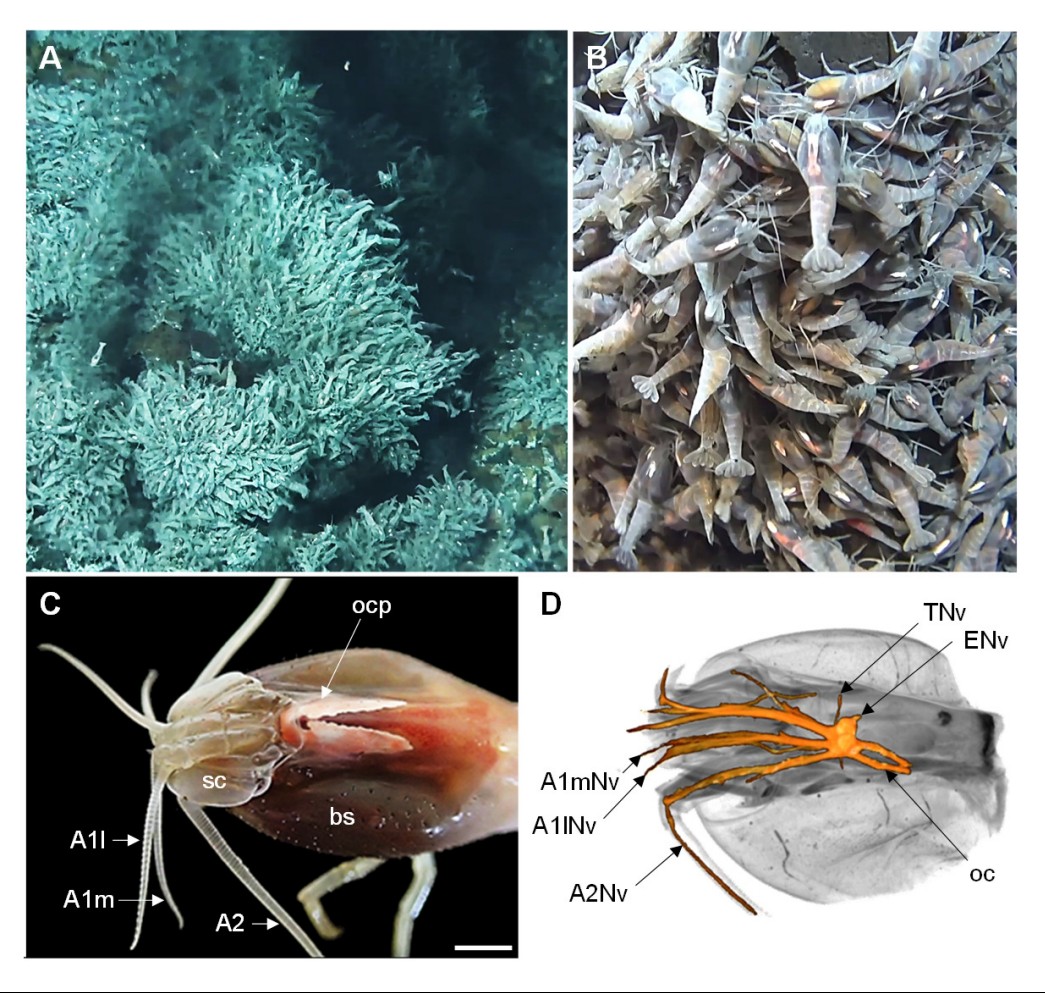

**Figure 1.** The Alvinocaridid vent shrimp *Rimicaris exoculata*. (**A,B**) Swarms of thousands of *R. exoculata* individuals are crowded along the walls of black smoker hydrothermal vents at the TAG vent site (3600 m depth), Mid-Atlantic Ridge (IFREMER/Nautile6000, BICOSE 2018 cruise). (**C**) Dorsal view of the cephalothorax of *R. exoculata*, showing voluminous gill chambers covered by the branchiostegites, dorsal eyes (i.e. ocular plate) with two elongated retinae fused in the anterior region, and sensory appendages (antennae 1 and 2). Scale bar = 5 mm. (**D**) Black-white inverted image from an X-ray micro-CT scan showing a dorsal overview of the *R. exoculata* cephalothorax, with 3D reconstruction of the brain and associated nerves. Abbreviations: see text and appendix 1.
DOI: https://doi.org/10.7554/eLife.47550.003

*Truchot, 1997*), or thermal stress (*Cottin et al., 2010*; *Ravaux et al., 2003*). However, the sensory mechanisms and adaptations used by the shrimps to perceive their habitat have only been partially investigated (see references below) despite their importance in understanding the lifestyle of vent shrimp species and their long-term evolution.

Vision and chemoreception have been proposed to be the major sensory modalities used by vent shrimp to perceive environmental cues (*Chamberlain, 2000*; *Jinks et al., 1998*; *Pelli and Chamberlin, 1989*; *Renninger et al., 1995*). In vent shrimps, the stalked compound eyes that characterize most malacostracan crustaceans are modified to form enlarged sessile eyes, which in *R. exoculata* are located underneath the dorsal carapace (*Chamberlain, 2000*; *Gaten et al., 1998*; *O'Neill et al., 1995*; *Van Dover et al., 1989*). The eyes cannot form images since the ommatidia lack a dioptric apparatus necessary to refract and focus rays of light, but the retina instead consists of hypertrophied rhabdoms and a reflective subjacent layer, structures that maximize the absorption of light.

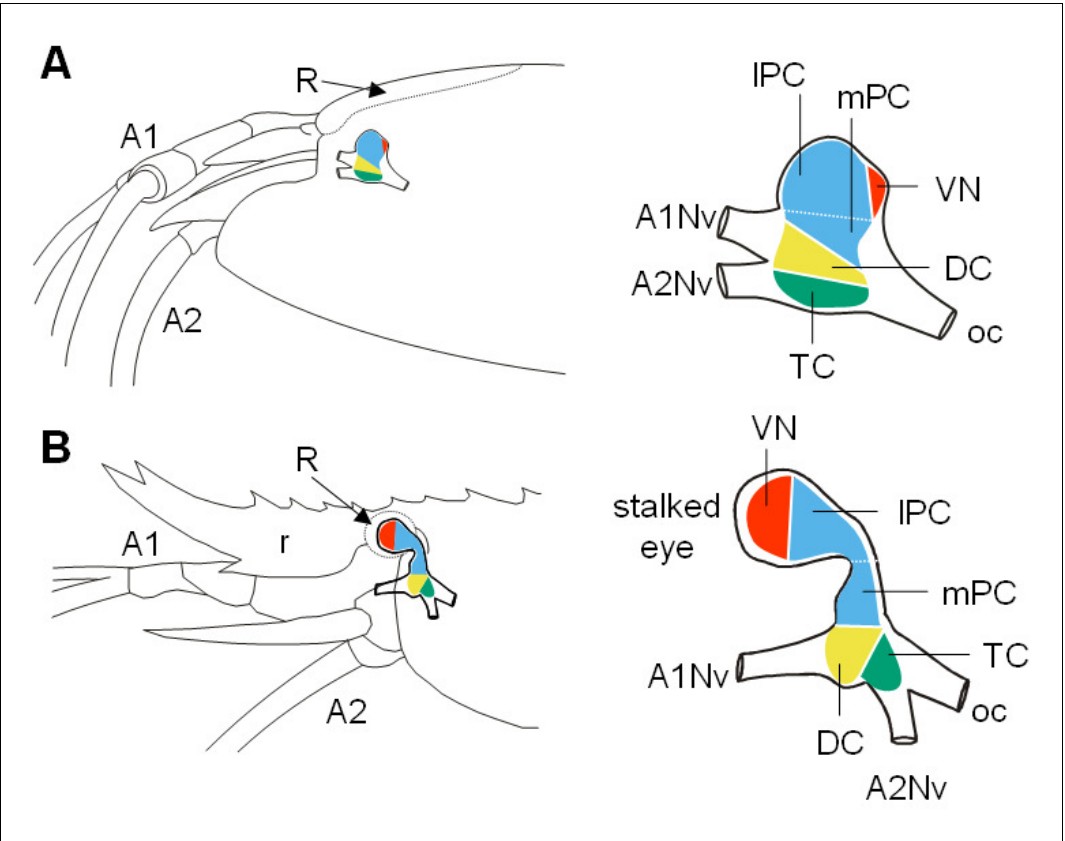

**Figure 2.** Comparative brain overview in Caridean vent and shallow-water species. Lateral sketches of the brains of the vent shrimp *Rimicaris exoculata* (**A**) and the closely-related shallow-water shrimp *Palaemon elegans* (**B**), showing the brain position within the cephalothorax, the position of the main nerves and the subdivision of the brain into three neuromeres called proto-, deuto- and tritocerebrum, plus the visual neuropils. In contrast to *P. elegans*, *R. exoculata* does not possess eyestalks and the visual neuropils are fused to the median brain, in a dorsoposterior position behind the lateral protocerebrum. Abbreviations: see text and appendix 1.
DOI: https://doi.org/10.7554/eLife.47550.004

These anatomical features could represent an adaptation to detect very dim light sources. It was suggested that the animals may perceive the black body radiation emitted by the extremely hot fluid which exits the chimney (*Chamberlain, 2000*; *Pelli and Chamberlin, 1989*; *Van Dover et al., 1989*). Furthermore, the animal's antennal appendages respond to sulfide, suggesting that vent shrimps can detect key chemical components of the hydrothermal fluid (*Machon et al., 2018*; *Renninger et al., 1995*), but sulfide detection is not restricted to vent shrimps since antennal responses were also recorded from shallow-water palaemonid shrimp (*Machon et al., 2018*). From structural descriptions of their antennae 1 and 2 and chemosensory sensilla, it is not clear whether their chemosensory system presents specific adaptations related to the hydrothermal environment (*Machon et al., 2018*; *Zbinden et al., 2017*). One specific feature of these organisms is the dense coverage of their antennal appendages by bacterial communities (*Zbinden et al., 2018*), whose potential roles remain unknown. Nevertheless, their occurrence on the sensory organs suggests a functional significance for the shrimp sensory abilities (*Zbinden et al., 2018*).

Crustacean brain structure is best understood in crayfish, crabs, and clawed and spiny lobsters (reviews for example *Derby and Weissburg, 2014b*; *Harzsch and Krieger, 2018*; *Schmidt, 2016*). We are interested in exploring adaptive changes of crustacean brain structures that have occurred during their evolutionary radiation into particular habitats and their adoption of specific life styles (e.g. *Harzsch et al., 2011*; *Kenning and Harzsch, 2013*; *Krieger et al., 2015*; *Krieger et al., 2012b*; *Krieger et al., 2010*; *Meth et al., 2017*). Differential investment in certain brain neuropils might reflect the sensory landscape which a certain crustacean species typically exploits, so that

studying an animal's brain anatomy may allow for predictions related to its ecology and lifestyle (*Sandeman et al., 2014*). For example, in peracarid and remipedian cave crustaceans, the visual neuropils are absent whereas the central olfactory pathway is well developed, highlighting that these blind animals may rely on olfaction as a major sensory modality in their lightless habitat (*Fanenbruck et al., 2004*; *Fanenbruck and Harzsch, 2005*; *Stegner et al., 2015*; *Stemme and Harzsch, 2016*). In representatives of the genus *Penaeus*, the olfactory system is moderately developed, while sophisticated antenna two neuropils are present, suggesting that the detection of hydrodynamic stimuli is important for these animals (*Meth et al., 2017*; *Sandeman et al., 1993*). Hence, comparing the architecture of the sensory centers among divergent crustacean lineages, across wide evolutionary distances and across diverse life styles, can help to understand structural adaptations to specific sensory environments (review in *Sandeman et al., 2014*). Studying crustaceans from extreme habitats is particularly informative in this respect (*Ramm and Scholtz, 2017*; *Stegner et al., 2015*). However, the structure of the brain in vent shrimps remains poorly understood (*Charmantier-Daures and Segonzac, 1998*; *Gaten et al., 1998*). Therefore, the present study sets out to provide a detailed description of the architecture of the *R. exoculata* brain against the background of the extreme conditions that characterize its habitat, and to ultimately discuss its contribution for crustacean brain evolution.

## Results

### Gross morphology of the cephalothorax

The wide cephalothorax of *Rimicaris exoculata* displays large branchiostegites (*bs*) which surround voluminous gill chambers (*Figure 1C*). The animals do not possess eyestalks but rather have bilaterally paired, wing-shaped eyes with a conspicuous, whitish retina that is fused in the anterior region to form the ocular plate (*ocp*). The lateral parts of the eye extend further dorsally and towards the posterior region of the cephalothorax (*Figures 1C* and *2A*). The first pair of antennae (*A1*) is biramous, with two flagella of similar length (*Figure 1C*). The second pair of antennae (*A2*) consists of a basal element, the scaphocerite (*sc*), and a long uniramous flagellum, slightly wider than those of the antennae 1 (*Figure 1C*). Micro-CT scans show that the brain is located in the anterior region of the cephalothorax, and receives main sensory afferences from the antenna 1 (*A1Nv*) and antenna 2 (*A2Nv*) nerves anteriorly, from the eye nerves (*ENv*) posterodorsally, from the tegumentary nerves (*TNv*) laterally, and from the oesophageal connectives (*oc*) posteriorly (*Figure 1D*).

### Overview of the brain architecture

Decapod crustacean brains are subdivided into three successive neuromeres: proto-, deuto- and tritocerebrum. In *R. exoculata*, these regions form a single, medially located mass (i.e. the median brain) (*Figures 1D* and *2A*). The visual neuropils are closely associated with the lateral protocerebrum (*lPC*), at a posterodorsal position (*Figure 2A*). This arrangement contrasts with other shallow-water carideans and most decapod crustaceans (see for example *Cronin and Porter, 2008*; *Meth et al., 2017*), in which the lateral protocerebrum is located at some distance from the median brain, in movable eyestalks (*Figure 2B*). The deutocerebrum (*DC*) is associated with the antenna one nerves, and the tritocerebrum (*TC*) is associated with the antenna two nerves (*Figure 2*). The brain's neuraxis is not aligned with body axis in *R. exoculata*, but is bent dorsally so that the protocerebrum is situated posterodorsally to the deutocerebrum (*Figure 2A*).

Data from micro-CT scans and aligned serial paraffin sections provided a consistent picture of the brain anatomy that we compiled in both three-dimensional reconstructions (*Figures 3A* and *4A–D*) and a schematic drawing of the *R. exoculata* brain (*Figure 3B,C*). In the following, for simplicity only one brain hemisphere is described, although mirror symmetrical structures are present in the contralateral hemisphere.

### Lateral protocerebrum: the visual neuropils

*R. exoculata* presents three successive visual neuropils, which are the lamina (*La*), medulla (*Me*), and lobula (*Lo*), from distal to proximal (*Figures 3*, *4* and *5*). The lamina is thin, flattened and elongated dorsally (*Figures 4A,D* and *5B-D*). The cell cluster (1) dorsally covers the lamina (*Figures 3C*, *4B, D* and *5C*). Numerous axon bundles from the entire length of the retina (*ENv*) converge onto the

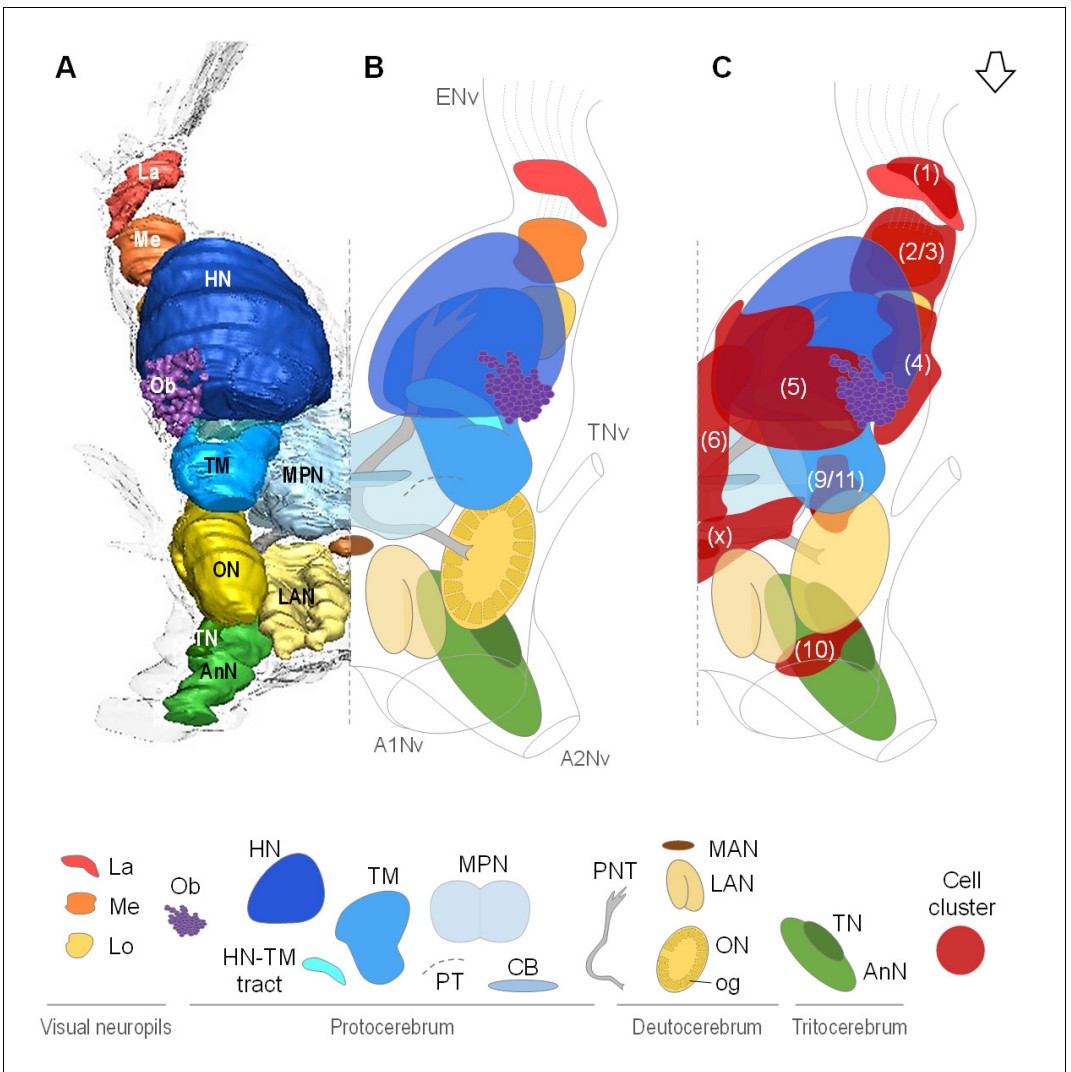

**Figure 3.** Overall organization of the brain of *R.exoculata*. 3D reconstruction (**A**) and schematic representations (**B**, **C**) of the brain and neuropils of *R. exoculata* viewed from a dorsal, slightly anterior direction. The open white arrow points towards anterior of the body axis. In (**C**), the clusters of cell somata associated with the neuropils are shown. The 3D reconstruction is based on an image stack obtained by serial sectioning of paraffin-embedded material; the sections were aligned manually by shifting and rotating each section using Amira. Abbreviations: see text and appendix 1.

DOI: https://doi.org/10.7554/eLife.47550.005

lamina (*Figures 3*, *4A–D* and *5D*). The retina consists of photoreceptor organelles, the rhabdoms (*dR*) (for which the degradation is ascribed to the damaging exposure to intense light during sampling and manipulation of the specimens at the surface), which overlie a white layer of reflecting cells, the tapetum (*T*), and clusters of pigment cells (*pc*) (*Figure 5D*, Figure 11A; *Nuckley et al., 1996*; *O'Neill et al., 1995*). The medulla is spherical (*Figures 3*, *4A–D* and *5B,C*) and is connected by thin fibers to the lamina (*Figure 5C*), the lobula (*Figure 5B*) and by a dense fibers tract to the terminal medulla (*Figure 5B*, *white arrowhead*). The lobula is slightly larger than the medulla, and is adjacent to the posterior side of the terminal medulla (*TM*) (*Figures 3*, *4A–D,F*, *5A* and *11A*). The merged cell clusters (2) and (3) cannot be clearly separated and cover both the medulla and the lobula (*Figures 3C*, *4B,D* and *5A–C*).

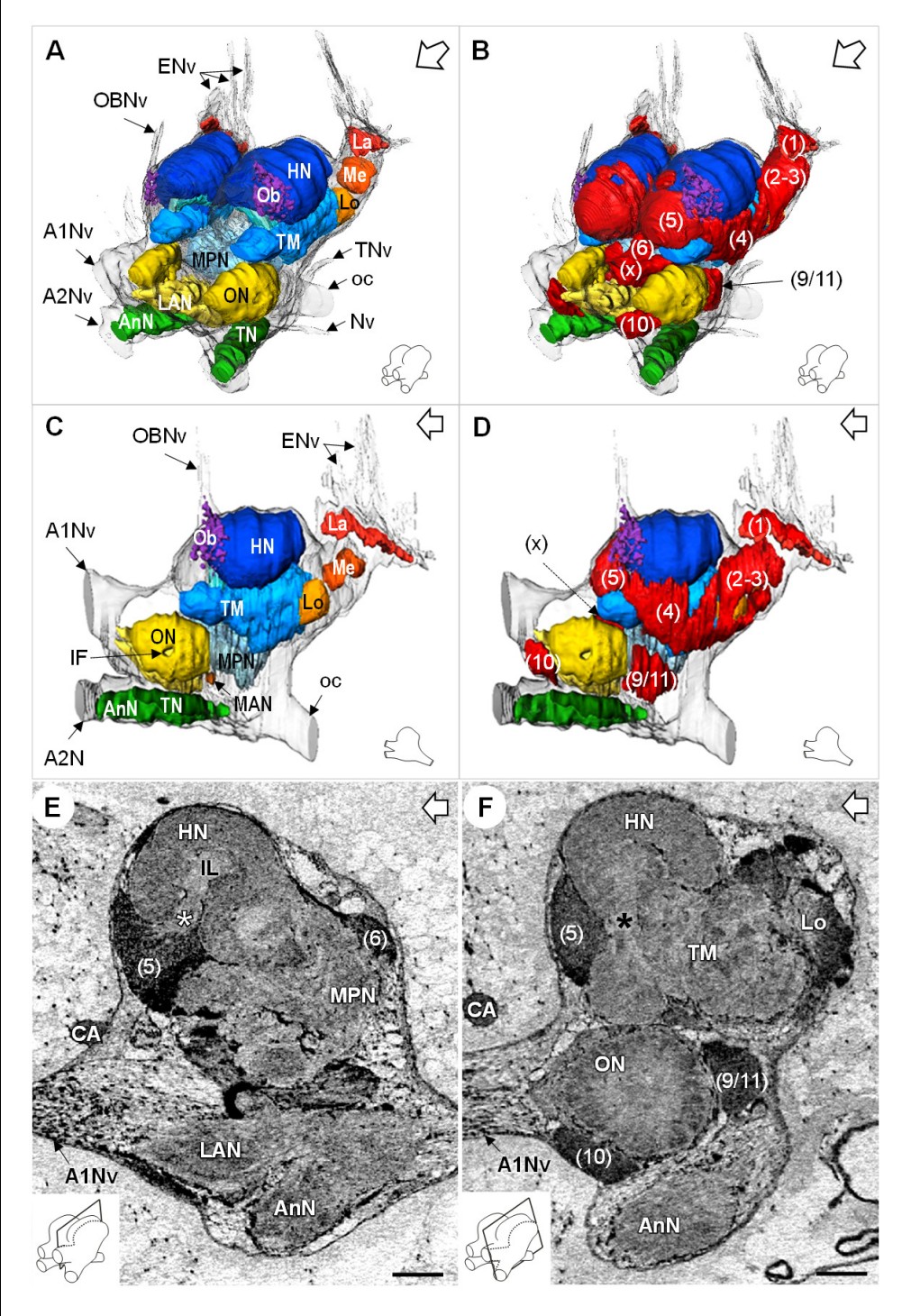

**Figure 4.** Additional views of the brain morphology in *R.exoculata*. (A–D) 3D reconstruction of the brain of *R. exoculata* in anterior-left (**A,B**) and left (**C, D**) views, based on an image stack obtained by serial sectioning of paraffin-embedded material. (**B** and **D**) include the cell clusters. The brain orientation is sketched in the bottom right corners. (**E,F**) Lateral sections of the brain of *R. exoculata* from micro-CT scans (black-white inverted images). The section's positions are depicted in the bottom left corners. White asterisk in (**E**) indicates the entrance of axons from the cell cluster (5) into the hemiellipsoid body. Black asterisk in (**F**) indicates the tract connecting the anterior region of the terminal medulla to the hemiellipsoid body. The open white arrows point towards anterior of the body axis. Scale bars = 100 μm. Abbreviations: see text and appendix 1.

DOI: https://doi.org/10.7554/eLife.47550.006

**Table 1.** Comparative table summarizing characteristics of aesthetascs and olfactory neuropils in several malacostracan species.

| Species (body length) | Aesthetascs | | Olfactory neuropils (ON) | | | |
| | Total number | Length (μm) | Neuropil total volume (x10^6 μm3) | Mean glomerular volume (x10^3 μm3) | Glomerular number | References |
|---|---|---|---|---|---|---|
| Leptostraca | | | | | | |
| *Nebalia herbstii* (1.4 cm) | - | - | 0.1 | 2 | 60 | *Kenning et al., 2013* |
| Stomatopoda | | | | | | |
| *Neogonodactylus oerstedii* (4 cm) | 80 | 400 | - | 110 | 70 | *Derby et al., 2003* |
| Isopoda | | | | | | |
| *Saduria entomon* (8 cm) | 40–60 | 240 | 3 | 34 | 80 | *Kenning and Harzsch, 2013*; *Pynnönen, 1985* |
| Dendrobranchiata | | | | | | |
| *Penaeus vannamei* (7 cm) | 280 | - | - | - | <100 | *Wittfoth and Harzsch, 2018*; *Zeng et al., 2002* |
| Caridea | | | | | | |
| *Palaemon elegans* (7 cm) | 280 | 230 | 120 | 225 | 530 | *Zbinden et al., 2017*; this study* |
| *Rimicaris exoculata* (6 cm) | 206 | 170 | 56 | 155 | 370 | *Zbinden et al., 2017*; this study |
| Achelata | | | | | | |
| *Panulirus argus* (20–60 cm) | 3000 | 1000 | 154 | 118 | 1332 | *Beltz et al., 2003*; *Grünert and Ache, 1988* |
| Homarida | | | | | | |
| *Homarus americanus* (20–60 cm) | 2000 | 600 | 141 | 592 | 249 | *Beltz et al., 2003*; *Guenther and Atema, 1998* |
| Astacida | | | | | | |
| *Procambarus clarkii* (9 cm) | 133 | - | 10 | 20 | 503 | *Beltz et al., 2003* |
| Anomura | | | | | | |
| *Birgus latro* (20 cm) | 1700 | - | 375 | 280 | 1338 | *Krieger et al., 2010* |
| *Coenobita clypeatus* (6 cm) | 519 | - | 120 | 154 | 799 | *Beltz et al., 2003* |
| *Pagurus bernhardus* (3 cm) | 673 | - | - | 171 | 536 | *Tuchina et al., 2015* |
| Brachyura | | | | | | |
| *Carcinus maenas* (9 cm) | 200 | 750 | - | 247 | - | *Fontaine et al., 1982*; *Hallberg and Skog, 2011* |

Estimates of the animal's body lengths are given for comparison. Carapace width is given for *B. latro* and *C. maenas*, and total length is given for all other species.

* The palaemonid shrimp *Palaemon elegans* was investigated in the present study for comparison, as a species closely-related to *R. exoculata* among the Caridea family.

DOI: https://doi.org/10.7554/eLife.47550.013

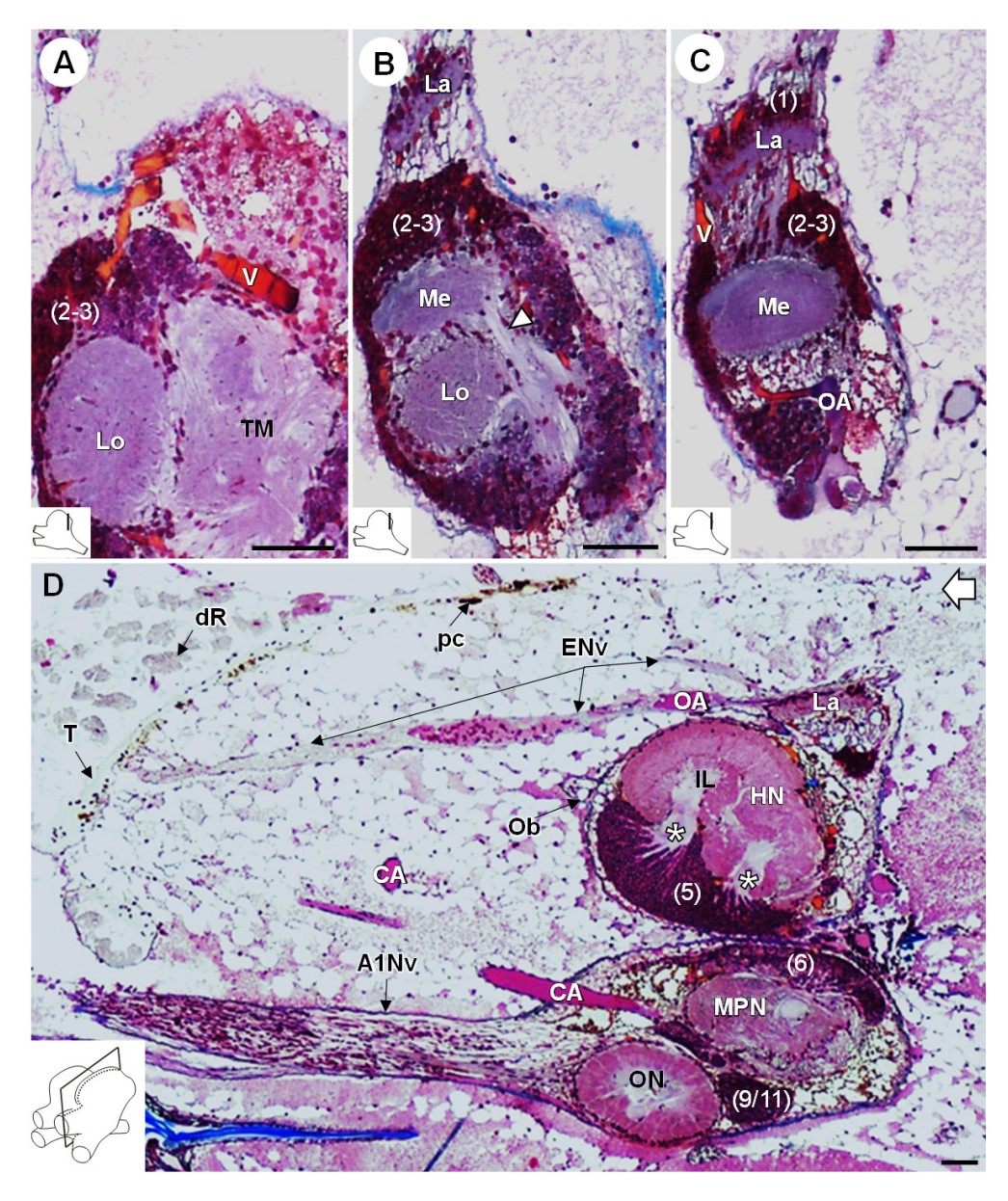

**Figure 5.** Lateral protocerebrum: the visual neuropils. (**A–C**) Frontal histological sections in the posterior region of the brain, from anterior to posterior, showing the visual neuropils, associated cell clusters, and part of the vascular system. The white arrow head in (**B**) shows the fiber tract connecting the medulla to the terminal medulla. (**D**) Sagittal histological section of the brain, showing the eye nerve fibers projecting from the anterodorsal retina to the lamina. White asterisks indicate the entrance of axons from the cell cluster (5) into the hemiellipsoid body intermediate layer. The open white arrow points towards anterior of the body axis. The section's positions are sketched in the bottom left corners. Scale bars = 100 µm. Abbreviations: see text and appendix 1.
DOI: https://doi.org/10.7554/eLife.47550.007

## Lateral protocerebrum: the hemiellipsoid body and terminal medulla

The lateral protocerebrum dominates the *R. exoculata* brain, with the hemiellipsoid body (*HN*), the terminal medulla (*TM*), together with the cell clusters (4) and (5) representing about 25% of the brain volume. The hemiellipsoid body is well defined, with a voluminous, hemispherical cap region (*HN_{cap}*) located dorsally (*Figures 3*, *4*, *5D* and *6A–F*) and displaying synapsin-like immunoreactivity (SYNir) (*Figure 6G–I*). The core region of the hemiellipsoid body (*HN_{core}*) is fused posteriorly with the

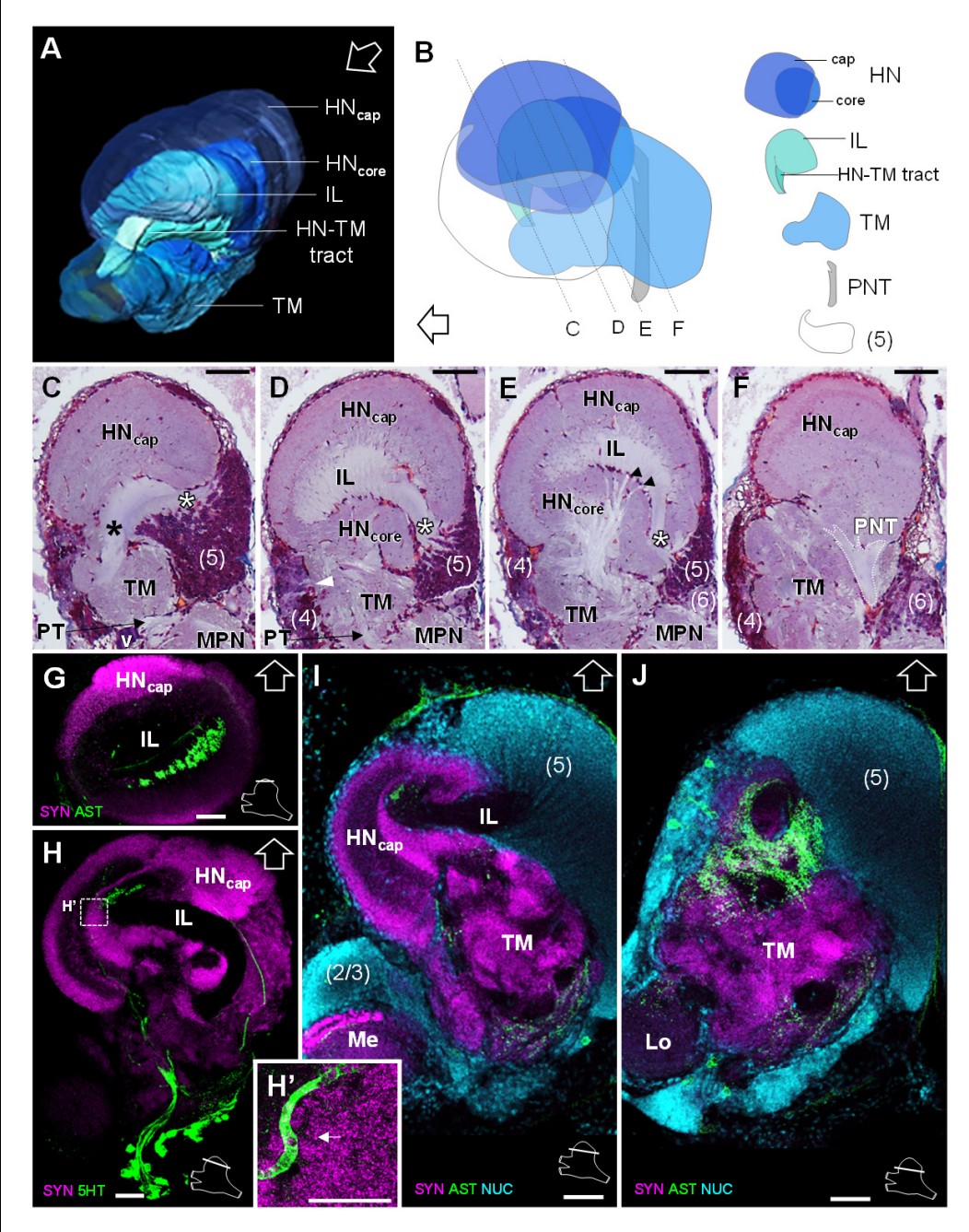

**Figure 6.** Lateral protocerebrum: the hemiellipsoid body and the terminal medulla. (A) 3D reconstruction of the lateral protocerebrum (right hemisphere), viewed from an anterior-left perspective, based on an image stack obtained by X-ray micro-CT scan. A conspicuous arcuate tract connects the anterior region of the terminal medulla to the cap region of the hemiellipsoid body (see also in C). (B) Schematic representation of the lateral protocerebrum (right hemisphere), viewed from the left. Dotted lines indicate the section's position in C-F. (C-F) Frontal histological sections of the lateral protocerebrum, from anterior to posterior. The hemiellipsoid body and the terminal medulla receive axons from the cell somata in the cell cluster (5) (*white asterisks*, (C–E) and (4) (*white arrowhead*, (D). An arcuate tract connects the terminal medulla to the cap region of the hemiellipsoid body in the anterior region (*black asterisk*, (C). The terminal medulla also connects to the hemiellipsoid body in the middle region, *via* arborizing fibers (*black arrowheads*, (E). The projection neuron tract enters the hemiellipsoid body in the posterior region (F). (G–J') Horizontal sections of the lateral protocerebrum, from dorsal to ventral, double or triple-labeled for synapsin immunoreactivity (SYN, *magenta*), allatostatin-like immunoreactivity (AST) or serotonin immunoreactivity (5HT) (both showed in *green*), and a nuclear marker (NUC, *cyan*). The inset (H') shows an

*Figure 6 continued on next page*

*Figure 6 continued*

enlargement of the hemiellipsoid body neuropil cap region, with microglomeruli (*white arrow*). Each section's position is sketched in the bottom right corners. Black and white open arrows point towards anterior of the body axis. Scale bars = 100 µm (except in H', scale bar = 50 µm). Abbreviations: see text and appendix 1.
DOI: https://doi.org/10.7554/eLife.47550.008

terminal medulla (*Figures 4F* and *6B,D,E*). The cap and core regions are separated by an arcuate intermediate layer (*IL*) (*Figures 4E* and *6A,B,D,E,G–I*) which receives parallel afferent fibers from the terminal medulla anteriorly (namely the HN-TM tract) (*Figures 3A,B*, *4F* and *6A–C*, *black asterisks*) and a massive bundle of neurites from somata in the cell cluster (5) at the medial side (*Figures 4E*, *5D* and *6C–E*, *white asterisks*). Some of the intermediate layer fibers display allatostatin-like immunoreactivity (ASTir) near the cap region (*Figure 6G*). The intermediate layer is devoid of SYNir (*Figure 6J*). The cap region is characterized by synaptic sites forming microglomeruli (*Figure 6H'*) and is also innervated by serotonergic neurons (*Figure 6H*). The cell cluster (5) is voluminous (*Figures 3C*, *4B* and *6B–E,I,J*) and contains approximately 30,000 cell somata of the so-called globuli cells (*Wolff et al., 2017*). The hemiellipsoid body receives input from the olfactory neuropils *via* the projection neuron tract (*PNT*) in the posterior region (*Figures 3A* and *6B,F*).

The terminal medulla is a large and complex neuropil. Anteriorly, it is shaped like a sphere (*Figures 3*, *4A,C,F* and *6A–C*), and it connects to the intermediate layer of the hemiellipsoid body *via* the HN-TM tract (*Figures 3A,D*, *4F* and *6A–C*, *black asterisks*). Posterior to this region, the terminal medulla is large, crossed by seemingly unstructured networks of fibers (*Figures 3A*, *4C,F* and *6D–F*) and displays SYNir (*Figure 6I,J*). It is innervated by neurites from the cell cluster (4) (*Figure 6D*, *white arrowhead*), and further connects again to the intermediate layer of the hemiellipsoid body *via* radiating fiber bundles (*Figure 6E*, *black arrowheads*).

## Median protocerebrum

The median protocerebrum (*mPC*) comprises two medially fused neuropils, the anterior (*AMPN*) and posterior (*PMPN*) medial protocerebral neuropils. The AMPN connects to the terminal medulla of the lateral protocerebrum anteriorly *via* the protocerebral tract (*PT*) (*Figures 3B* and *6C,D*) and the PMPN *via* the posterior protocerebral tract (*PPT*), the latter containing neurites with strong serotonin-immunoreactivity (5HTir) (*Figure 7C*) and seemingly interconnecting the terminal medulla of both hemispheres. Both, the AMPN and PMPN are separated by the unpaired central body neuropil (*CB*) (*Figure 3*), which displays ASTir (*Figure 7B*), weak SYNir (*Figure 7A*) and strong 5HTir (*Figure 7C*). Overall, the median protocerebrum contains many fibers from serotonergic neurons, partly from the cell cluster (x) (which likely refers to the cell clusters (12 , 13) and (17) according to *Sandeman et al., 1992*), which define well the elements of the central complex, that is the protocerebral bridge (*PB*) and the central body (*CB*), and also the posterior region associated to the posterior protocerebral tract. Posteriorly to the central body, fiber bundles of the projection neuron tracts from both hemispheres meet in a region with strong SYNir, that we will call the projection neuron tract central neuropil (PNTCN) (*Figure 7A*).

## Deutocerebrum

In the deutocerebrum, a paired neuropil with a conspicuous structure is located laterally, the lobe-shaped olfactory neuropil (*ON*) (*Figures 3*, *4A–D,F*, *5D* and *8A–F*). It is composed of approximately 180 wedge-shaped neuropil units, the olfactory glomeruli (*og*), which are arranged radially around the periphery of a non-synaptic core (*Figures 3B*, *4F*, *5D* and *8A–F*). Each glomerulus shows strong SYNir (*Figure 8E–F*), as well as ASTir which highlights a subdivision of each glomerulus into a cap (*c*), subcap (*sbc*) and base (*b*) region (*Figure 8E'*). The sensory input of the olfactory neuropil comes from the olfactory sensory neurons innervating the aesthetasc sensilla on the lateral flagellum of the antenna 1. The somata of olfactory interneurons located in the cell cluster (9/11) innervate fibers of the olfactory neuropil, some of which display ASTir (*Figure 8H*). These fibers enter *via* the medial foramen (*mF*) into the core of the neuropil (*Figure 8C,E*), from where they target the glomerular base region (*Figure 8E*), or cross to the lateral foramen (*lF*) (*Figure 8A,C,E*) to spread out laterally and innervate the glomerular cap region (*Figure 8E*, *white arrowhead*). The medial foramen is also

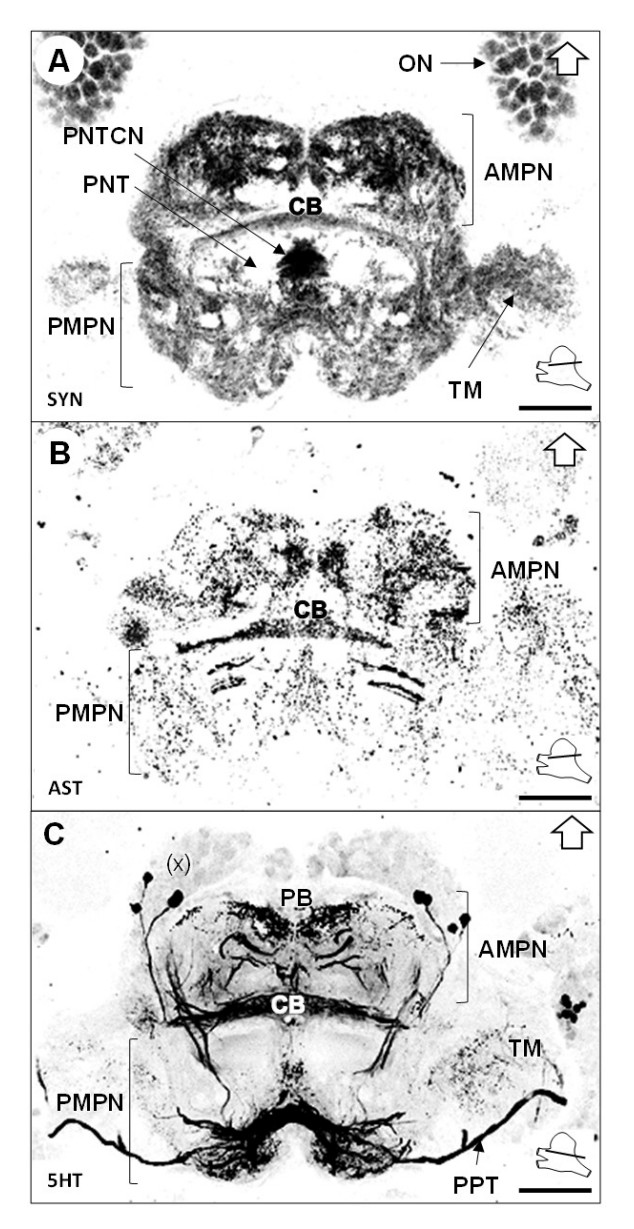

**Figure 7.** Median protocerebrum. (**A–C**) Black-white inverted images of horizontal sections of the median protocerebrum labeled for synapsin immunoreactivity (**A**), allatostatin-like immunoreactivity (**B**) or serotonin immunoreactivity (**C**). The section's positions are sketched in the bottom right corners. Black arrows point towards anterior of the body axis. Scale bars = 100 μm. Abbreviations: see text and appendix 1.
DOI: https://doi.org/10.7554/eLife.47550.009

the place where efferent fibers exit from the olfactory neuropil. These are the axons of the olfactory projection neurons that form the projection neuron tract (*Figure 8B,D*). A projection neuron tract neuropil (PNTN) as known from other decapods (e.g. *Sandeman et al., 1992*; *Harzsch and Hansson, 2008*; *Krieger et al., 2012a*) is identifiable close to the ascending branch of the tract (*Figure 8E,F*). The projection neuron tract then transverses the median protocerebrum and projects to the lateral protocerebrum (see above).

The lateral antenna one neuropil (*LAN*) is located medially to the olfactory neuropil. It is U-shaped (*Figures 3* and *8A,B,H*) and displays strong SYNir, as well as ASTir, which reveals a transversely stratified pattern (*Figure 8G*). This neuropil connects posterodorsally to the median protocerebrum

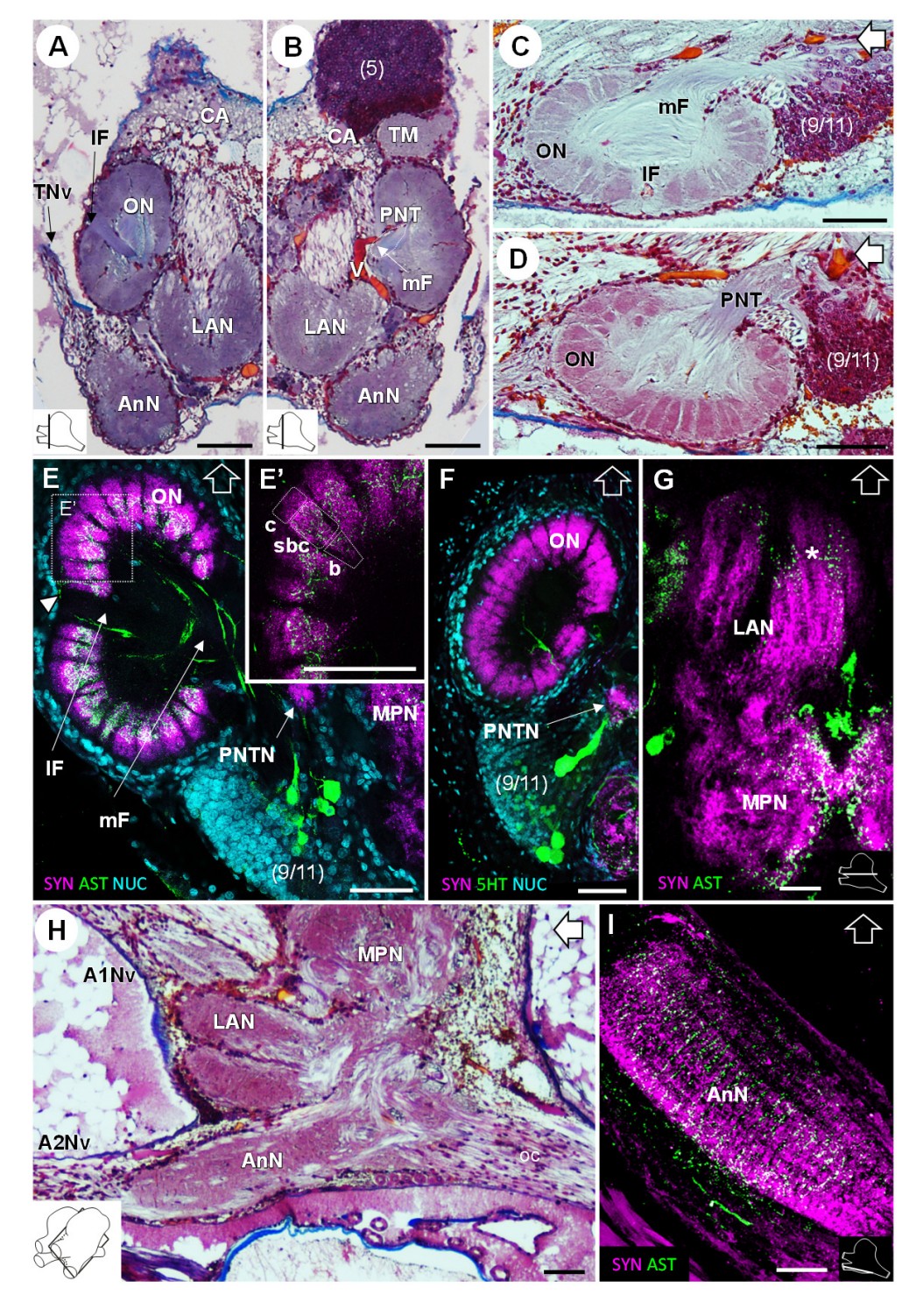

**Figure 8.** Deutocerebrum and tritocerebrum. (**A,B**) Overview of the deutocerebrum and tritocerebrum (frontal histological sections). (**A**) is anterior to B. (**C,D**) Sagittal histological sections of the olfactory neuropils. (**E,F**) Horizontal sections of the olfactory neuropil, triple-labeled for synapsin immunoreactivity (SYN, *magenta*), allatostatin-like immunoreactivity (AST, (**E,E'**) or serotonin immunoreactivity (5HT, (**F**) (*green*), and a nuclear marker (NUC, *cyan*). (**G**) Horizontal section of the transversely stratified (*white asterisk*) lateral antenna one neuropil, double-labeled for synapsin immunoreactivity (SYN, *magenta*) and allatostatin-like immunoreactivity (AST, *green*). (**H**) Sagittal histological section of the tritocerebrum, part of the deutocerebrum and median protocerebrum.

*Figure 8 continued on next page*

*Figure 8 continued*

(I) Horizontal section of the transversely stratified antenna two neuropil, double-labeled for synapsin immunoreactivity (SYN, *magenta*) and allatostatin-like immunoreactivity (AST, *green*). The section's positions are sketched in the bottom corners. Black and white open arrows point towards anterior of the body axis. Scale bars = 100 µm. Abbreviations: see text and appendix 1.
DOI: https://doi.org/10.7554/eLife.47550.010

(*Figure 8G,H*). The median antenna one neuropil (*MAN*) is small, poorly defined, and located in the center of the deutocerebrum, below the anterior region of the median protocerebrum and between the paired lateral antenna one neuropils (*Figure 3A,B*).

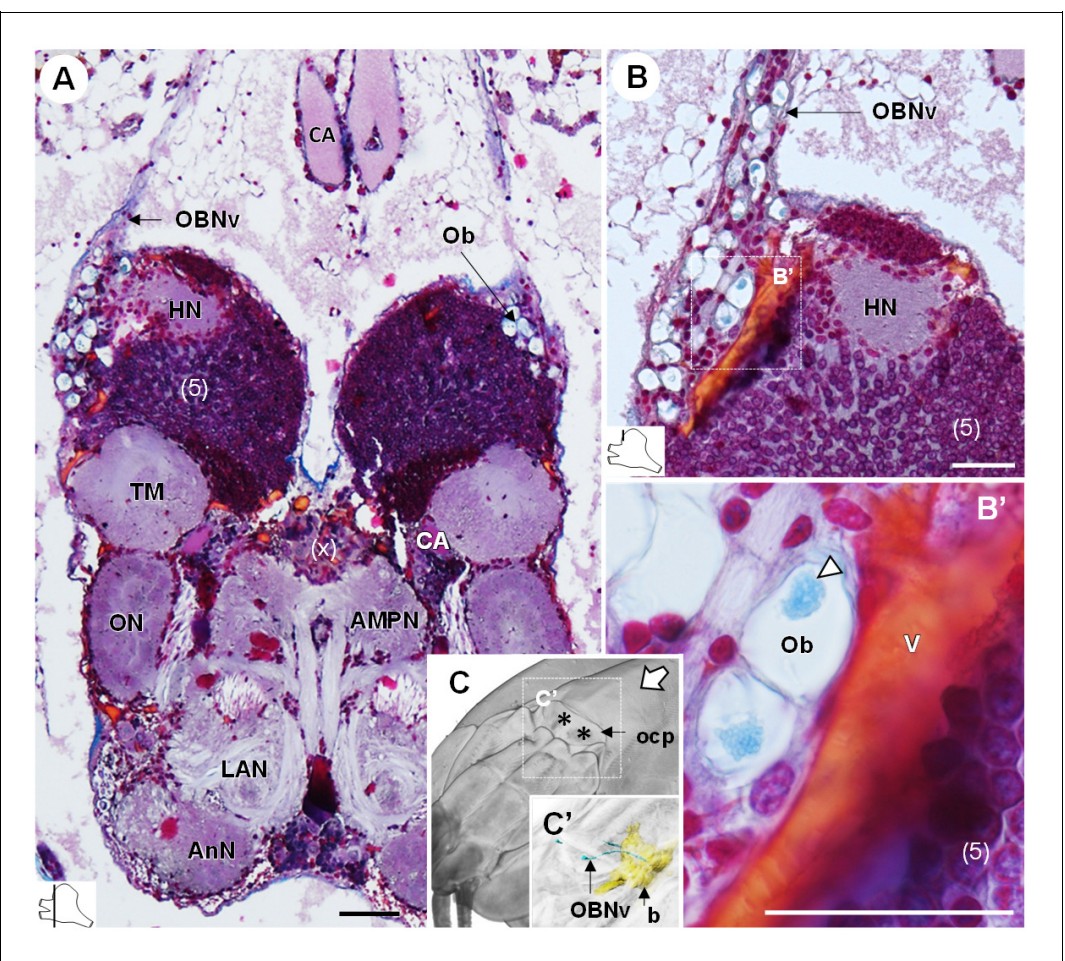

**Figure 9.** The organ of Bellonci. (A–B') Frontal histological sections of the anterior region of the brain, showing conspicuous onion body-structures from which a nerve tract emanates (A,B), and which are seemingly closely associated to the cerebral vascular system (B') and contain elements of granular appearance (B'), *white arrowhead*). The section's positions are sketched in the bottom left corners. (C) Anterodorsolateral overview of the cephalothorax from micro-CT scan. Asterisks indicate the position where the organ of Bellonci nerve connects to the cuticle beneath the anterior region of the ocular plate. (C') shows a 3D reconstruction of the brain and the organ of Bellonci nerve in this region. White arrow points towards anterior of the body axis. Scale bars = 100 µm. Abbreviations: see text and appendix 1.
DOI: https://doi.org/10.7554/eLife.47550.011

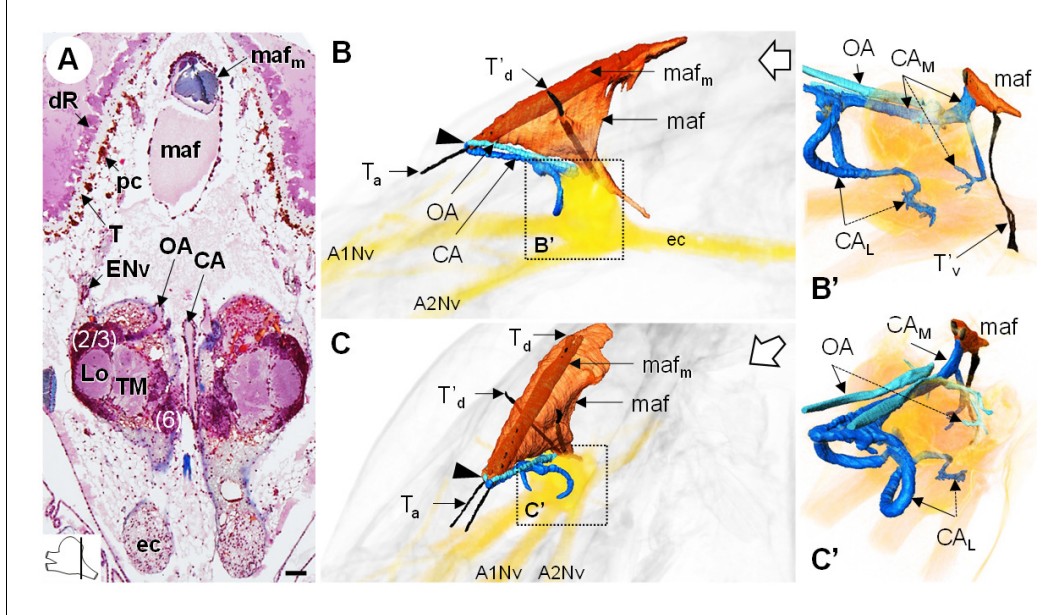

**Figure 10.** The myoarterial formation and cerebral vascular system. (**A**) Frontal histological section of the myoarterial formation located between the bilateral retina and above the visual neuropils. The section's position is sketched in the bottom left corner. Scale bar = 100 μm. (**B,C**) 3D reconstruction of the myoarterial formation (*orange*), part of the cerebral vascular system (*blue* and *cyan*) and the brain (*yellowish*), from lateral (**B**) and anterolateral (**C**) views, in the cephalothorax. (**B'** and **C'**) show higher magnifications of the cerebral vascular system. Dotted arrows indicate structures inside the brain. Open white arrows point towards anterior of the body axis. Abbreviations: see text and appendix 1.
DOI: https://doi.org/10.7554/eLife.47550.012

## Tritocerebrum

The tritocerebrum comprises the antenna two neuropil (*AnN*), which has a cylindrical shape and lies in front of the oesophageal connectives (*Figures 3* and *4*, **8** HA). SYNir and ASTir show a transversely stratified pattern within this neuropil (*Figure 8I*). Poorly differentiated from the antenna two neuropil, the tegumentary neuropil (*TN*) is located posterodorsally (*Figure 3*).

## The organ of Bellonci

The organ of Bellonci (*OB*) is typical for many crustaceans but its sensory function remains unclear (*Chaigneau, 1994*). In *R. exoculata*, this organ is conspicuous and comprises onion bodies (*Ob*) structures connected to a well-developed nerve tract (*OBNv*). The onion bodies are situated on the anterolateral side of the brain, in front of the hemiellipsoid body (*Figures 3*, *4A–D*, *5D* and *9A*). They represent a cluster of about fifty densely packed lobules (*Figure 9A,B*), many of them containing elements of granular appearance (*Figure 9B'*, *white arrowhead*). Some lobules are further located in the proximal region of OBNv (*Figure 9B*). This nerve is large in its proximal region, and progressively tapers as it draws away from the brain (*Figure 9A*). Anterodorsally, the nerve extends through the retinal layers and connects underneath the cuticle of the ocular plate (*Figure 9C,C'*).

## The myoarterial formation and cerebral vascular system

The myoarterial formation (*maf*) (or cor frontale, auxiliary heart) underlies the dorsal carapace and is located between the paired eyes, above the brain (*Figure 10A,B,C*). This organ is voluminous, being almost as long as the elongated retina, and extends ventrally towards the dorsal region of the brain (*Figure 10B',C'*). Two adjacent and parallel muscle bundles (*maf_m*) penetrate through the myoarterial formation (*Figure 10A*) and attach to the cuticle *via* tendons located either anteriorly ($T_a$) or dorsally ($T_d$) (*Figure 10B,C*). Two thinner muscular bundles cross the myoarterial formation in its middle region, perpendicular to the main adjacent muscles, and are attached both to the dorsal and ventral

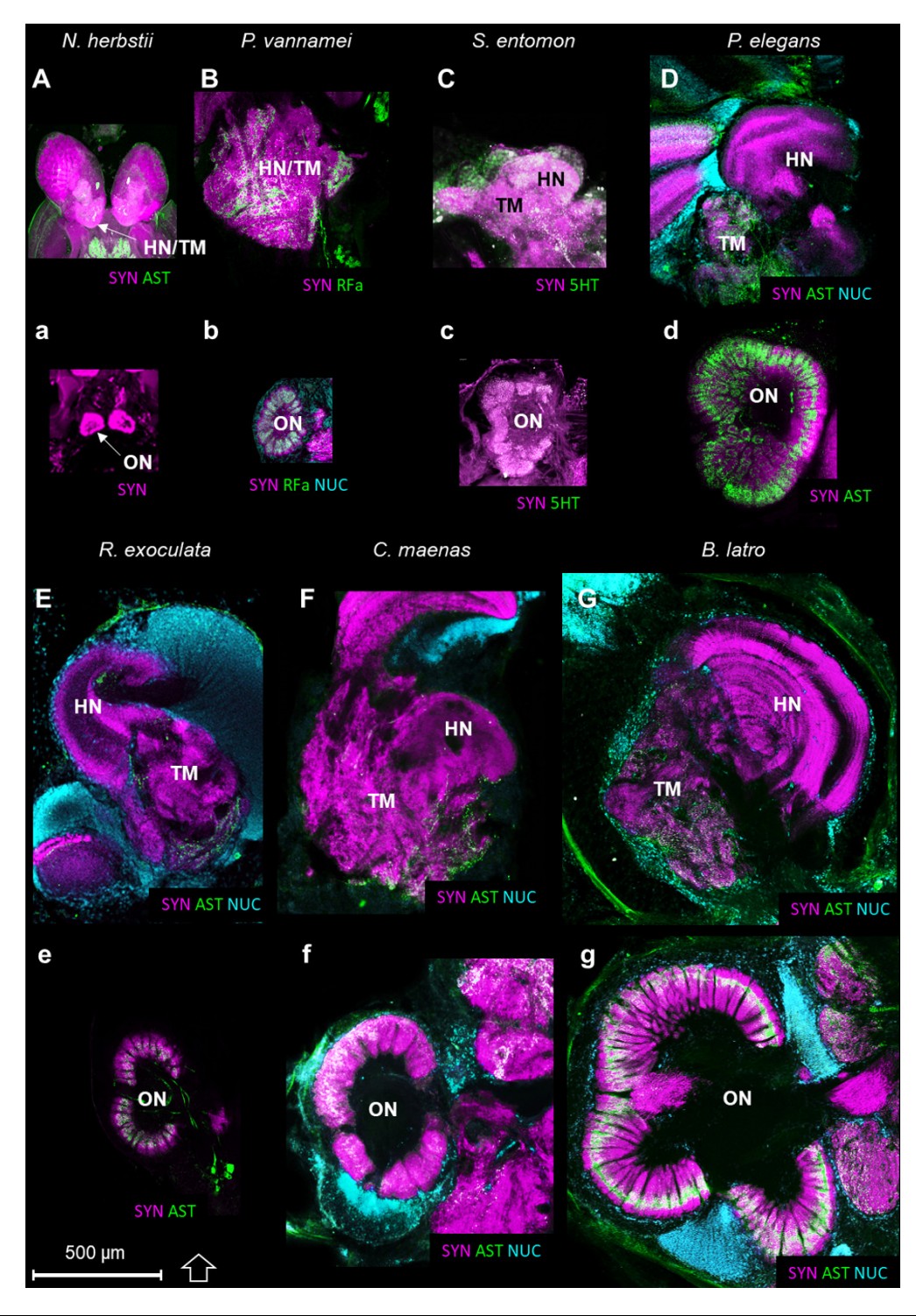

**Figure 11.** Comparison of the higher integrative centers and olfactory neuropils in several representatives of Malacostraca displayed at the same scale. Sections of the higher integrative centers (i.e. hemiellipsoid body and terminal medulla) (**A–G**) and horizontal sections of the olfactory neuropil (**a–g**) labeled with different sets of antibodies (see below), in several malacostracan species: *Nebalia herbstii* (**Aa**), Leptostraca, from *Kenning and Harzsch (2013)*; *Kenning et al. (2013)*, *Penaeus vannamei* (**Bb**), Dendrobranchiata, from *Meth et al. (2017)*, *Saduria entomon* (**Cc**), Isopoda, from *Kenning and Harzsch (2013)*, *Palaemon elegans* (**Dd**) and *Rimicaris exoculata* (**Ee**) (Caridea, this study), *Carcinus maenas* (**Ff**), Brachyura, from *Krieger et al. (2012b)* and *Birgus latro* *Figure 11 continued on next page*

*Figure 11 continued*

(*Gg*), Anomala, from **Krieger et al. (2010)**. Markers: a, SYNir; B, SYNir +RFair; b, SYNir +RFair + NUC; Cc), SYNir +5 HTir; A,d,e, SYNir +ASTir; D,E,F-g, SYNir +ASTir + NUC. *ASTir*, allatostatin-like immunoreactivity (*green*); *NUC*, nuclear marker (*cyan*); *RFair*, RFamide-like immunoreactivity (*green*); *SYNir*, synapsin immunoreactivity (*magenta*); *5HTir*, serotonin immunoreactivity (*green*). Abbreviations: see text and appendix 1.

DOI: https://doi.org/10.7554/eLife.47550.014

cuticle of the cephalothorax, by secondary dorsal and ventral tendons ($T'_d$, *Figure 10B,C*; $T'_v$, *Figure 10B'*).

Anteriorly, below the junction of the ocular plate with the dorsal carapace, the myoarterial formation gives rise to three conspicuous, large cerebral arteries, a central cerebral artery (*CA*) and two ophthalmic arteries (*OA*), which all make a steep U-turn and extend parallel towards the dorsal side of the brain (*Figure 10B,C*, *arrowhead*). In an anterodorsal position, median to the hemiellipsoid bodies, the central cerebral artery divides into three smaller arteries, one median ($CA_M$) and two lateral ones ($CA_L$) (*Figure 10B',C'*). The median artery passes over the brain, between the two spherical masses of the lateral protocerebrum, and then divides into two branches, one entering the brain posteriorly, at the level of the median protocerebrum, and a larger one merging with the ventral region of the myoarterial formation. This suggests a loop system wherein part of the hemolymph in the cerebral artery goes back into the myoarterial system. The lateral cerebral arteries are coil-shaped and enter the brain above the insertion of the antenna one nerve to target for instance the olfactory neuropils and the lateral antenna one neuropils in the deutocerebrum. The ophthalmic arteries enter the brain in a posterodorsal position (*Figure 10C'*) and target the visual neuropils and the lateral protocerebrum.

The cerebral vascular system of *R. exoculata* is considerably developed, with blood vessels supplying all brain neuropils and cell clusters, as in other crustaceans, including large vessels that irrigate the visual neuropils (*Figure 5A,C*), the deutocerebrum (*Figure 8B*) and the lateral protocerebrum (*Figure 9B,B'*). The Azan staining reveals pink-to-purple cerebral arteries that enter the brain (*CA*) (*Figures 5C,D*, *8A,B* and *10A*), and orange vessels inside the brain (*V*) (*Figures 5A,C*, *8B* and *9B*).

## Comparisons of the olfactory system and the higher integrative centers with other crustaceans

*Table 1* presents a comparison of aesthetasc and olfactory neuropil characteristics in different taxa of crustaceans. The number of olfactory glomeruli and their unitary volume in *R. exoculata* fit within the ranges displayed by other decapods. However, this species presents relatively small olfactory neuropils (excluding the fibrous core) compared to other species with roughly the same body size, such as the caridean *Palaemon elegans* and the anomuran *Coenobita clypeatus*. Yet, other species of about the same body size (e.g. the crayfish *Procambarus clarkii* and the isopod *Saduria entomon*) possess olfactory neuropils of even smaller volume than those of *R. exoculata*.

The higher integrative centers (i.e. the hemiellipsoid body and the terminal medulla, associated with cell cluster 5) are especially well-developed in *R. exoculata* in relation to the relative size of the olfactory neuropils, compared to other crustaceans (*Figure 11*). As an example, from relative volumes obtained from 3D reconstructions, the higher integrative centers in *R. exoculata* occupy approximately 25% of the total brain volume, similarly to the caridean shrimp *P. elegans* (22%) but twice more than in the giant robber crab *Birgus latro* (13%). However, the olfactory neuropils of *P. elegans* and *B. latro* occupy roughly two and six times more volume than those of *R. exoculata*, values that represent 4.2%, 17% and 2.7% of the total brain volume, respectively in these species.

## Discussion

### General remarks

The alvinocaridid shrimp *Rimicaris exoculata* is an endemic species to hydrothermal vent habitats, well adapted to these deep sea environments with peculiar physicochemical conditions. The present study sets out to gain insights into adaptations to specific features of the vent habitat (e.g. low ambient light levels and steep variations of chemical concentrations). The analysis of the brain

architecture in *R. exoculata* aims to highlight relative investments into certain neuronal subsystems, in relation with the animal's habitat and lifestyle. The general anatomy of the brain of *R. exoculata* corresponds in many aspects to the ground pattern of the malacostracan crustacean brain (*Kenning et al., 2013*), including the subdivision into proto-, deuto- and tritocerebrum, the location of main nerves and the presence of distinct cell clusters. However, the brain of *R. exoculata* also exhibits morphological differences to other malacostracans, especially at the level of the lateral protocerebrum. The organ of Bellonci is especially conspicuous (also observed by *Charmantier-Daures and Segonzac, 1998*), but its sensory function remains elusive. In the following, we will focus on the structure of major sensory centers (i.e. the visual system, the olfactory system and the higher integrative centers). We will also discuss the evolution of the hemiellipsoid bodies as higher integrative brain centers, which are substantial in *R. exoculata*. We will begin our account by addressing the neurovascular system that supplies the brain.

## The neurovascular system

In crustaceans, the neurovascular system has been described mainly in crayfish (*Chaves da Silva et al., 2013*; *Scholz et al., 2018*), crabs (*McGaw, 2005*; *McGaw and Reiber, 2002*; *Sandeman, 1967*) and spiny lobsters (*Steinacker, 1979*) (reviews in *McMahon, 2001*; *Steinacker, 1979*; *Steinacker, 1978*; *Wilkens, 1999*). The brain and eyes are supplied with hemolymph *via* the anterior aorta system, which originates antero-medially from the heart and runs between the stomach and the dorsal integument (*Scholz et al., 2018*). Anteriorly, a dilatation of the anterior aorta, the myoarterial formation (*Scholz et al., 2018*; also named the cor frontale in for example *McGaw (2005)*; *Steinacker (1978)* which functions as an auxiliary heart, pumps the hemolymph specifically towards the anterior part of the central nervous system. In malacostracan crustaceans, the myoarterial formation above the brain gives rise to a descending cerebral artery, which supplies the median brain, and to two ophthalmic arteries that turn laterally and extend into the eyestalks to supply the visual neuropils (*Chaves da Silva et al., 2013*; *McGaw, 2005*; *Scholz et al., 2018*). In *R. exoculata*, consistent with the absence of eyestalks, the myoarterial formation and its arteries differ in shape, size and position from those previously described in other malacostracans (*Figure 10*). Among the potential corollaries for the pronounced neurovascular system in *R. exoculata*, one is the more efficient hemolymph pumping to the brain. In crustaceans, the perfusion of the brain is modulated by physiological or environmental factors, such as hypoxia (*Reiber and McMahon, 1998*). Because the pure hydrothermal fluid is anoxic, the mixing of the fluid with the surrounding seawater can create hypoxic conditions for vent animals (*Childress and Fisher, 1992*; *Schmidt et al., 2008*). Known adaptations to hypoxia in vent crustaceans include an hemocyanin with a higher affinity for oxygen compared to shallow-water species (*Chausson et al., 2004*; *Lallier and Truchot, 1997*; *Sanders et al., 1988*). A very pronounced capillary network was also observed in hydrothermal vent alvinellid polychaetes (*Hourdez and Lallier, 2006*). Accordingly, the pronounced myoarterial formation and large cerebral arteries in *R. exoculata* could represent a particularly efficient system for oxygen delivery to the brain to cope with low availability of oxygen.

## A visual system adapted to a dim light environment

*Van Dover et al. (1989)* described the *R. exoculata* eyes as a pair of large anteriorly fused organs that underlie the transparent dorsal carapace of the cephalothorax and demonstrated the presence of rhodopsin-like visual pigments in high quantity, with a maximum absorption at 500 nm. Subsequent analyses showed that the eyes comprise a smooth cornea located above a dense layer of hypertrophied rhabdoms, under which a white layer of reflective cells, the tapetum, is located and maximizes the absorption of light by the photoreceptors (*Chamberlain, 2000*; *Jinks et al., 1998*; *Nuckley et al., 1996*; *O'Neill et al., 1995*). These elements of the retina were all discernible in our histological sections (*Figure 5D*), although the rhabdoms were strongly degenerated, a process ascribed to the damaging exposure to intense light during sampling (*Herring et al., 1999*; *Johnson et al., 1995*). The eyes of *R. exoculata* lack the dioptric apparatus which characterizes the ommatidia of compounds eye of pelagic and shallow water crustaceans and thus cannot form images (*Chamberlain, 2000*; *Jinks et al., 1998*; *Nuckley et al., 1996*; *O'Neill et al., 1995*), but their highly sensitive naked retina seems adapted for the detection of low ambient light levels, to the detriment of spatial resolution (*Chamberlain, 2000*; *Van Dover et al., 1989*).

In malacostracan crustaceans, the visual input from the compound eyes is processed by a suite of retinotopic visual neuropils, usually but not exclusively located within the moveable eyestalks (*Figure 2B*) (*Strausfeld, 2012*; *Loesel et al., 2013*). The absence of eyestalks of *R. exoculata* coincides with a strong size reduction and fusion of the visual neuropils with the median brain (*Figure 2A*). Nevertheless, already *Charmantier-Daures and Segonzac (1998)* and *Gaten et al. (1998)* differentiated three visual neuropils in *R. exoculata*, namely the lamina, medulla, and lobula, as in the ground pattern of the Malacostraca. However, in *R. exoculata* these neuropils are located posterodorsally to the enlarged lateral protocerebrum (*Figures 2A* and *4A,C*). The dorsal expansion of the flattened lamina, that extends in parallel to the retina, may indicate a retinotopic projection of photoreceptor input onto the lamina which could allow the animals to extract directional information from light sources above but this issue must be addressed in future experiments. In the medulla, immunohistochemistry revealed an outer layer (*Figure 6I*) (which is also faintly visible in histological sections, *Figure 5B,C*), suggesting a subdivision of the medulla into an outer and inner region, as seen in crayfish (*Strausfeld and Nassel, 1981*). No such stratification was observed for the lobula (*Figure 5A,B*), and synapsin immunoreactivity was weak in this most proximal neuropil (*Figure 6J*), although in malacostracans with well-developed compound eyes, the lobula displays numerous, neurochemically diverse strata (e.g. Brachyura and Anomura, *Harzsch and Hansson, 2008*; *Krieger et al., 2012b*; *Krieger et al., 2010*; *Wolff et al., 2012*; *Polanska et al., 2007*; *Meth et al., 2017*; *Strausfeld, 2005*). The simplified structure of the lobula, which in other malacostracans plays a role in motion detection (*Strausfeld, 2012*), might mirror the inability of the eye to form images. Also, the lobula plate, a fourth visual neuropil present in several malacostracan taxa (e.g. *Bengochea et al., 2018*; *Harzsch and Hansson, 2008*; *Krieger et al., 2012a*; *Krieger et al., 2010*; *Sztarker et al., 2009*; *Meth et al., 2017*; *Strausfeld, 2005*; *Kenning et al., 2013*; *Kenning and Harzsch, 2013*; *Sinakevitch et al., 2003*) could not be identified in *R. exoculata*. The lobula plate has been suggested to mediate optokinetic control, necessary to track moving objects (e.g. conspecifics, preys, predators) (*Sztarker et al., 2005*). Such a role is consistent with the loss of the lobula plate in *R. exoculata*, which lacks the realization of image formation, necessary for tracking moving objects.

Many eyeless representatives of Crustacea have partially or totally lost their central visual pathways while adapting to a life under dim-light conditions or complete darkness (e.g. *Ramm and Scholtz, 2017*; *Stegner et al., 2015*; *Elofsson and Hessler, 1990*; *Stegner and Richter, 2011*; *Fanenbruck et al., 2004*; *Brenneis and Richter, 2010*). It is likely that the reduction of these nervous tissues is promoted under the selective pressure of those conditions resulting in a less energy expenditure for organisms living in constant or partial darkness, since eliminating neuronal structures which are no longer useful saves considerable amounts of energy (*Klaus et al., 2013*; *Moran et al., 2015*; *Niven and Laughlin, 2008*).

However, the fact that neuronal elements indicative for a functional visual system are present in *R. exoculata* must mean that there is light to exploit as an environmental cue. Also, the unusual nature of the visual system of *R. exoculata* suggests that it exploits a specific type of signal. One prominent hypothesis refers to the thermal black body radiation emitted by the hot hydrothermal fluid at the chimney's exit with a temperature of up to 350°C, which peaks in the infrared but part of its spectrum extends into the visible light (*Pelli and Chamberlin, 1989*; *Van Dover et al., 1996*; *Van Dover et al., 1988*; *Van Dover and Fry, 1994*). The ability to localize this radiation could serve both to attract the shrimp to optimal areas for supplying its symbionts with vital, reduced compounds of the hydrothermal fluid, and to allow avoidance of scorching fluid (*Van Dover et al., 1989*). Visual cues other than thermal radiation are likely to be also exploited by *R. exoculata*, related to turbulence, mixing and precipitation, such as chemi-, crystallo-, tribo- and sono-luminescence, for which the emission spectra lie between 450–800 nm (*Reynolds and Lutz, 2001*; *Tapley et al., 1999*; *Van Dover et al., 1996*; *Van Dover and Fry, 1994*; *White, 2000*; *White et al., 2002*).

## The olfactory system

Two modes of chemoreception, linked to distinct chemosensory pathways, are distinguished in malacostracan crustaceans (*Derby and Weissburg, 2014b*; *Schmidt and Mellon, 2010*): olfaction, which is mediated by the aesthetasc sensilla located on the lateral flagellum of the antenna 1, and distributed chemoreception, which is mediated by the bimodal chemo- and mechanosensory sensilla

located mainly on all antennal appendages, the mouthparts, and the walking appendages (*Garm et al., 2005*; *Garm et al., 2003*; *Garm and Watling, 2013*; *Mellon, 2014*; *Mellon, 2012*; *Schmidt and Gnatzy, 1984*). *R. exoculata* presents aesthetascs in similar number and dimensions to other caridean representatives (*Table 1*), as well as several bimodal sensilla with different morphologies on the antennal appendages (*Zbinden et al., 2017*).

Olfaction has been extensively studied in malacostracans (e.g. *Ache, 2002*; *Derby and Weissburg, 2014b*; *Schmidt and Mellon, 2010*), and the central olfactory pathway has received much attention in crustacean neuroanatomy (e.g. *Blaustein et al., 1988*; *Harzsch and Krieger, 2018*; *Kenning et al., 2013*; *Kenning and Harzsch, 2013*; *Krieger et al., 2015*; *Krieger et al., 2012b*; *Krieger et al., 2010*; *Sandeman et al., 1992*; *Schachtner et al., 2005*; *Schmidt and Mellon, 2010*). The afferent olfactory input from the olfactory sensory neurons innervating the aesthetascs targets the conspicuous olfactory neuropils, which are lobe-shaped and bilaterally arranged in the deutocerebrum (*Figure 8A–F*). They are composed of spherical or cone-shaped dense synaptic neuropils, namely the olfactory glomeruli, which are radially arranged around the periphery of a core of non-synaptic fibers. The olfactory glomeruli are subdivided into a cap, subcap and base regions in several decapod taxa (e.g. *Harzsch and Krieger, 2018*; *Schachtner et al., 2005*; *Schmidt and Ache, 1997*). The glomeruli of *R. exoculata* appear to conform to this design principle, with an identical subdivision (*Figure 8E'*). Although the number of olfactory glomeruli is roughly in the same range to that of its close relative *Palaemon elegans* of about the same body size, the olfactory neuropils of *R. exoculata* are relatively small in terms of volume compared to *P. elegans* (*Table 1*). Notably, the olfactory neuropils of *R. exoculata* are moderately developed compared to other species (*Table 1* and *Figure 11*). Hence, the dimensions and structural complexity of the olfactory neuropils in *R. exoculata* do not suggest, judging from comparative brain anatomy, that the loss of the eye's capacity to form images is compensated by sophisticated olfactory abilities.

Efficient olfactory abilities would have been especially relevant to probe the chemical environment of *R. exoculata*, which is dynamic, with strong concentration variations of hydrothermal fluid chemicals as the hydrothermal fluid dilutes with the surrounding seawater. Sulfide and other chemicals could serve as highly important environmental cues for *R. exoculata* (*Renninger et al., 1995*; *Machon et al., 2018*) to locate active edifices as optimal areas to supply its chemoautotrophic symbionts with reduced compounds. However, sulfide detection is likely mediated by distributed chemoreception, or both distributed chemoreception and olfaction, rather than exclusively olfaction, as it can be detected by the flagella of the antenna two which does not bear aesthetascs (*Machon et al., 2018*). Olfaction is also involved in the recognition of conspecifics (*Breithaupt and Thiel, 2011*) and the localization of sexual partners (*Wyatt, 2014*), but there is to date no detailed information on the inter-individual interactions in and out of the swarms of *R. exoculata*. The detection of chemical cues produced by bacteria could also appear especially relevant since the sensory antennal appendages of vent shrimp are often covered by a dense bacterial layer, whose roles are currently unknown (*Zbinden et al., 2018*).

## Evolution of higher integrative brain centers: the hemiellipsoid body

Malacostracan crustaceans display a rich repertoire of complex behavioral patterns related to finding food, shelter and mating partners, kin recognition and brood care, as well as orientation and homing. Decapod crustaceans are also known for complex social interactions such as communal defensive tactics, the occupation of common shelters, cooperative behavior during long-distance, offshore seasonal migration and the establishment of dominance hierarchies (*Breithaupt and Thiel, 2011*; *Derby and Thiel, 2014a*; *Duffy and Thiel, 2007*; *Thiel and Watling, 2015*). Because such complex behaviors most likely involve elements of learning and memory, higher integrative brain centers are suggested to provide the neuronal substrate for more sophisticated processing underlying such behaviors (review in *Sandeman et al., 2014*). Such centers receive input exclusively from second or higher order neurons but not from any primary sensory afferents (i.e. from the peripheral nervous system) and contain interneurons responding to the stimulation of several different sensory systems. In the malacostracan brain, the (bilaterally paired) terminal medulla, hemiellipsoid body, and accessory lobe seem to function as higher integrative centers, all three distinct neuropil areas which display a high level of complexity and are notable for their substantial volume (*Sandeman et al., 2014*). The terminal medulla and the closely associated hemiellipsoid body, are targeted by axons of the olfactory projection neurons as output pathway of the olfactory neuropil and accessory lobe (where

present; reviews *Derby and Weissburg, 2014b*; *Harzsch and Krieger, 2018*; *Schmidt, 2016*). Because of these anatomical relationships, evolutionary (*Sullivan and Beltz, 2004*; *Sullivan and Beltz, 2001*) and functional considerations (*Harzsch and Krieger, 2018*; *Sandeman et al., 2014*; *Strausfeld, 2012*) have focused on possible roles of these centers in higher order olfactory processing. In addition to the olfactory projection neuron axons, the terminal medulla also receives input from the visual neuropils in several malacostracans (reviewed in *Sandeman et al., 2014*). A specific type of local interneuron associated to the medulla terminalis and the hemiellipsoid body are the parasol cells (*Mellon and Alones, 1997*; *McKinzie et al., 2003*; *Mellon et al., 1992a*; *Mellon, 2000*; *Mellon et al., 1992b*) which respond to olfactory, tactile, and visual stimuli, thus highlighting their role as elements in higher order integration (*Mellon and Alones, 1997*; *Mellon, 2003*; *Mellon, 2000*; *Mellon and Wheeler, 1999*). Recent evidence obtained from a brachyuran crab suggests an involvement of the crustacean hemiellipsoid body/terminal medulla complex in memory processes (*Maza et al., 2016*). Furthermore, considering anatomical similarities of the crustacean hemiellipsoid body and insect mushroom body, *Wolff et al. (2017)* suggested an involvement in place memory.

During the evolutionary elaboration of malacostracan brains, substantial modifications occurred related to the relative proportion of types of input and investment in size of the various higher integrative centers (*Figure 11*; *Harzsch and Krieger, 2018*; *Sandeman et al., 2014*). Because the terminal medulla has a highly complex and highly variable structure, being composed of several, partly confluent neuropil lobes with heterogeneous appearance containing both coarse and fine fibers (e.g. *Blaustein et al., 1988*), its architecture so far has not been studied in a comparative context. We will focus in the following on the hemiellipsoid body whose structure is somewhat easier to grasp (*Figure 11*). In its simplest form, the hemiellipsoid body consists of a volume of fine neuropil with little texture that is closely associated with the terminal medulla (*Figures 11A,B,C,F* and *12*). Such a phenotype is for example common in leptostracans, the presumably most basal branch of the Malacostraca (*Figures 11A* and *12*; *Kenning et al., 2013*), but also in representatives of the Dendrobranchiata (*Figures 11B* and *12*; *Meth et al., 2017*; *Sullivan and Beltz, 2004*), and several Brachyura (*Figures 11F* and *12*; *Krieger et al., 2010*; *Krieger et al., 2012a*; *Krieger et al., 2015*). Isopoda as representatives of the Peracarida also feature simple, dome-shaped hemiellipsoid bodies (*Figures 11C* and *12*; *Kenning and Harzsch, 2013*; *Stemme and Harzsch, 2016*) whereas in Amphipoda (*Ramm and Scholtz, 2017*) and blind groups of peracarids from relict habitats (*Stegner et al., 2015*), this center is poorly developed and may be entirely missing (*Figure 12*). A more complex phenotype features a separation of the hemiellipsoid body into two separated areas, an architecture present for example in the spiny lobsters (neuropils I and *Blaustein et al., 1988*), the crayfish *Procambarus clarkii* and *Orconectes rusticus* (neuropil I and II, *Sullivan and Beltz, 2001*), and *Cherax destructor* (*Sullivan and Beltz, 2005*) (*Figure 12*). The clawed lobster *Homarus americanus* also features two neuropil units, but these are stacked on top of each other as cap and core neuropils separated by an intermediate, non-synaptic layer (*Sullivan and Beltz, 2001*) (*Figure 12*). Additional differences exist between the crayfish and the clawed lobster concerning the areas that are targeted by the axons of the projection neurons. (*Mellon et al., 1992a*; *Mellon et al., 1992a*; *Sullivan and Beltz, 2005*; *Sullivan and Beltz, 2001*). Hemiellipsoid bodies with a cap/core structure separated by an intermediate layer are also present in the brains of marine (*Krieger et al., 2012a*) and terrestrial hermit crabs of the taxon Coenobitidae, *Coenobita clypeatus* (*Harzsch and Hansson, 2008*; *Polanska et al., 2012*; *Wolff et al., 2012*) and *Birgus latro* (*Krieger et al., 2010*). These animals all display a large hemiellipsoid body with a peripheral, dome-shaped cap neuropil enclosing two dome-shaped core neuropil areas Core one and Core 2 (*Figures 11G* and *12*). Their hemiellipsoid body is associated with several thousands of small, intrinsic neurons (*Harzsch and Hansson, 2008*; *Krieger et al., 2010*). The cap and core neuropils are separated by intermediate layers formed by the neurites of these intrinsic interneurons and the afferents of the projection neuron tract in a rectilinear arrangement (*Wolff et al., 2012*).

In the hemiellipsoid bodies of the stomatopod crustaceans *Gonodactylus bredenii* (*Sullivan and Beltz, 2004*) and *Neogonodactylus oerstedii* (*Wolff et al., 2017*), the cap/core motif is modified such that the cap layer (termed 'calyx' in *Wolff et al., 2017*) is much thinner than the core neuropil (*Figure 12*) and that the cluster of intrinsic neurons expands over much of the surface of the cap neuropil. The additional stalked neuropils in the lateral protocerebrum of *N. oerstedii* (*Wolff et al., 2017*) will not be discussed here for simplicity. In *Stenopus hispidus* (Stenopodidea), the hemiellipsoid body appears very complex in structure, with apparently three distinct lobular neuropils

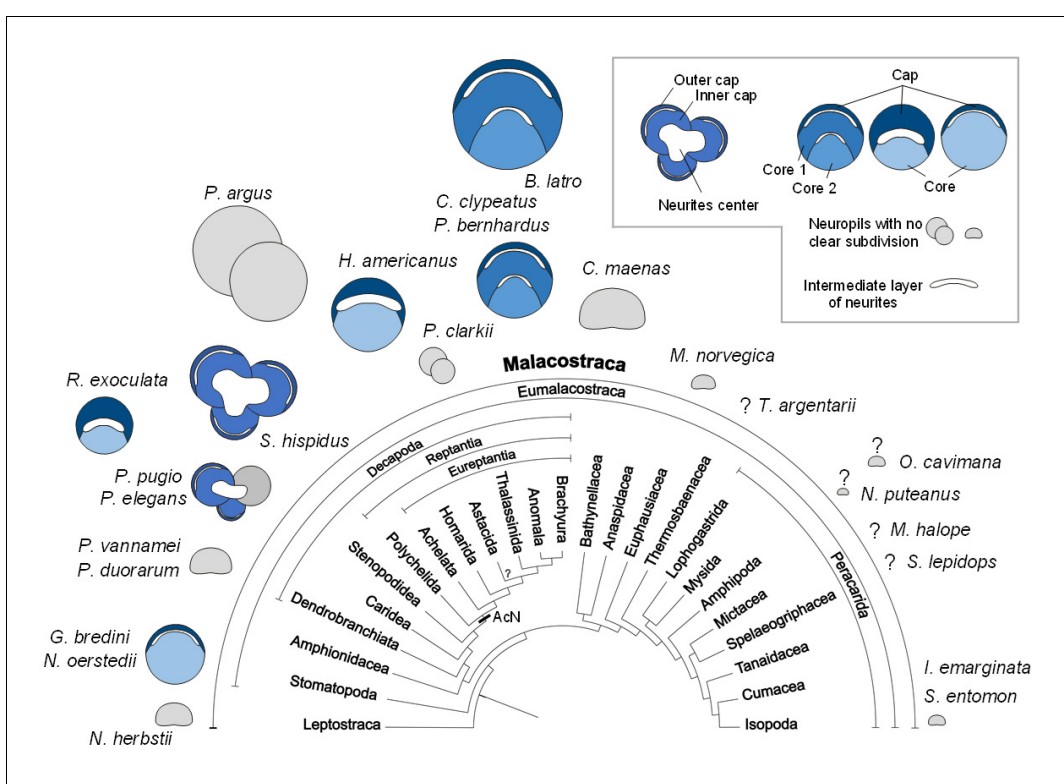

**Figure 12.** Structure of the hemiellipsoid body in several representatives of Malacostraca. The sketches of the hemiellipsoid body structure are displayed in relative size and include representatives of Leptostraca (*Nebalia herbstii*, **Kenning et al., 2013**), Stomatopoda (*Neogonodactylus oerstedii*, **Wolff et al., 2017**; *Gonodactylus bredini*, **Sullivan and Beltz, 2004**), Dendrobranchiata (*Penaeus vannamei*, **Meth et al., 2017**; *Penaeus duorarum*, **Sullivan and Beltz, 2004**), Caridea (*Rimicaris exoculata* and *Palaemon elegans*, this study; *Palaemonetes pugio*, **Sullivan and Beltz, 2004**), Stenopodidea (*Stenopus hispidus*, **Sullivan and Beltz, 2004** and Krieger et al. unpublished), Achelata (*Panulirus argus*, **Blaustein et al., 1988**), Homarida (*Homarus americanus*, **Sullivan and Beltz, 2001**), Astacida (*Procambarus clarkii*, **Sullivan and Beltz, 2001**), Anomala (*Birgus latro*, **Krieger et al., 2010**; *Coenobita clypeatus*, **Wolff et al., 2012**); *Pagurus bernhardus*, **Krieger et al., 2012a**), Brachyura (*Carcinus maenas*, **Krieger et al., 2012a**), Euphausiacea (*Meganyctiphanes norvegica*, unpublished), Thermosbaenacea (*Tethysbaena argentarii*, **Stegner et al., 2015**), Amphipoda (*Orchestia cavimana* and *Niphargus puteanus*, **Ramm and Scholtz, 2017**), Mictacea (*Mictocaris halope*, **Stegner et al., 2015**), Spelaeogriphacea (*Spelaeogriphus lepidops*, **Stegner et al., 2015**) and Isopoda (*Saduria entomon*, **Kenning and Harzsch, 2013**; *Idotea emarginata*, **Stemme et al., 2014**). Sketches were made from sections stained using antibody raised against synapsin, except *N. puteanus* (antibody raised against tubulin), *M. norvegica*, *P. argus* and *O. cavimana* (histological sections), and *N. herbstii* (optical section). The symbol '?" indicates that the presence of a hemiellipsoid body is uncertain. The phylogram showing phylogenetic relationships of malacostracan crustaceans is modified from **Harzsch and Krieger (2018)** (therein modified from **Sandeman et al., 2014**, as compiled after **Richter and Scholtz, 2001**; **Scholtz and Richter, 1995**; **Wirkner and Richter, 2010**). Abbreviations: see text and appendix 1.

DOI: https://doi.org/10.7554/eLife.47550.015

(*Figure 12*; *Sullivan and Beltz, 2004*; Krieger et al., unpublished). The hemiellipsoid body in the caridean species *P. elegans* and *Palaemonetes pugio* also presents three lobular neuropils, two of which present a cap layer and one or two core regions, and a third neuropil without clear subdivision (*Figures 11D* and *12*; *Sullivan and Beltz, 2004*). The hemiellipsoid body of *R. exoculata* in many aspects, closely corresponds to the cap/core layout (*Figures 6*, *11E* and *12*) although it is slightly simpler than in Coenobitidae with only one core neuropil, similar to the arrangement observed in *H. americanus* (*Sullivan and Beltz, 2001*) (*Figure 12*).

In summary, the hemiellipsoid body displays more structural variations across the Malacostraca than many other elements of the crustacean brain areas which led *Sandeman et al. (2014)* to note

that, within the Malacostraca, several different evolutionary trajectories are present to increase their brain's capacity for integrating olfactory and multimodal stimuli. This diversity masks common motifs of hemiellipsoid body architecture, explaining why genealogical relationships of the crustacean and insect protocerebral multimodal centers have been discussed controversially for many years (reviews e.g. *Loesel et al., 2013*; *Sandeman et al., 2014*; *Strausfeld, 2012*; *Strausfeld, 2009*; *Strausfeld, 1998*). Recent evidence suggests that, despite many morphological differences, these protocerebral structures of insects and crustaceans nevertheless share common architectural, physiological and neurochemical features suggesting a homology of their very basic neuronal circuitry (*Brown and Wolff, 2012*; *Maza et al., 2016*; *Wolff et al., 2012*; *Wolff et al., 2017*; *Wolff and Strausfeld, 2015*).

## Possible functions of the hemiellipsoid body: new lesson from *R. exoculata*?

Because the projection neuron tract provides a massive input to the lateral protocerebrum, recent comparative considerations have suggested that the structural elaboration and size of hemiellipsoid bodies largely mirror the importance of the central olfactory pathway in a given brain, thus emphasizing their role in higher order olfactory processing (*Harzsch and Krieger, 2018*; *Sandeman et al., 2014*). Along these lines, *Harzsch and Hansson (2008)* and *Krieger et al. (2010)* noted that in representatives of the Coenobitidae, the architectural complexity and volume of the olfactory neuropil closely correlates to that of the hemiellipsoid body. The comparative plates (*Figures 11* and *12*) demonstrate that *R. exoculata* dramatically deviates from this pattern in that their disproportionally large hemiellipsoid body contrasts with inconspicuous and moderately developed olfactory neuropils. The observation that visual input is likely to also play a subordinate role in these animals compared to shallow-water relatives with fully developed compound eyes makes us suggest that in *R. exoculata*, their impressive hemiellipsoid body may fulfill functions in addition of higher order sensory processing. Although the size alone does not qualify to be better performing (*Chittka and Niven, 2009*) and caution must be taken to conclude about functional differences from differences in size of structures (*Striedter, 2005*), comparative anatomical studies may lead to functional hypotheses. Discussing anatomical similarities of the crustacean hemiellipsoid body and insect mushroom body, *Wolff et al. (2017)* suggested for these two neuropils a role in place memory, based on observations that insects with elaborate navigational skills display elaborate mushroom bodies. Considering recent experiments that suggest an involvement of the crustacean hemiellipsoid body/terminal medulla-complex in memory processes (*Maza et al., 2016*), we here propose that the hemiellipsoid body in *R. exoculata* is involved in the formation of place memory. This hypothesis is further supported by the presence of serotonergic tracts within the hemiellipsoid bodies (*Figure 6H*), since serotonin has a function for place memory and learning in the mushroom bodies of *Drosophila melanogaster* (e.g. *Sitaraman et al., 2008*). Spiny lobsters *Panulirus argus* are renowned for their extensive offshore migrations and their ability to orient accurately towards their home sites over long distances by using the direction of water movement (surge) caused by wave action, learned local structural features, and geomagnetic cues for navigation (reviewed in *Sandeman et al., 2014*). Using a GPS-based telemetric system, giant robber crabs, *Birgus latro*, were shown to form route memories and may use path integration as navigation strategy and in translocation experiments were shown to be capable of homing over large distances (*Krieger et al., 2012b*). The above mentioned crustacean species display hemiellipsoid bodies impressive in size or structure. For survival in the extreme, lightless habitat of *R. exoculata*, an excellent place memory may be essential for avoiding the dangerously hot vent chimneys and memorizing emission sites of hydrothermal fluids rich in those chemicals on which their endosymbiont bacteria depend.

## Conclusion

Our observations of the general brain architecture of *R. exoculata* highlight several unusual characteristics, which could be related to adaptations to the specific sensory landscape of the vent habitat. The well-developed neurovascular system could be particularly efficient for brain oxygenation, to cope with the low availability of oxygen in the close surroundings of active chimneys. The conservation of the visual pathway and neuropils in a mostly aphotic environment suggests that vision nevertheless is a relevant sense for vent shrimp. The olfactory system does not present unusual traits and

olfaction is probably not a dominant sensory modality in this shrimp unlike what has been proposed so far (*Renninger et al., 1995*). On the other hand, the higher integrative centers are well-developed. The hemiellipsoid bodies are disproportionally large relative to the visual and olfactory neuropils size, and could be involved in complex integrative processes such as place memory. Overall, vent shrimp appear to be especially interesting models to investigate both sensory adaptations to extreme environmental conditions, and the evolution of the sensory centers among Crustacea.

## Materials and methods

### Animal collection and fixation procedures

Specimens of alvinocaridid shrimp *R. exoculata* (*Williams and Rona, 1986*) were collected on the TAG vent site (MAR, 26˚08'N-44˚49'W, 3600 m depth) during the BICOSE 2018 cruise on the Research Vessel 'Pourquoi Pas?'. Animals were sampled with the suction device of the Diving Support Vessel 'Nautile 6000', and recovered at their in situ pressure using the PERISCOP isobaric recovery device (*Shillito et al., 2008*). Immediately after retrieval, specimens were dissected to remove the hepatopancreas prior to fixation. The specimens for histology and x-ray micro-computed tomography (*micro-CT*) scans were stored in Bouin's fixative (10% formaldehyde, 5% glacial acetic acid in saturated aqueous picrinic acid) at 4˚C until use. The specimens for immunohistochemistry were fixed 24 to 48 hr in 4% formaldehyde (*FA*) in 0.1 M Phosphate buffered saline (*PBS*) at 4˚C for 24 hr, and then stored in 0.1 M PBS with $NaN_3$ at 4˚C until use. All specimens were sexed using the sexual dimorphism from the second pair of pleopods. Specimens are females for all micro-CT and histology experiments, and are both females and males for immunohistochemistry.

Caridean shallow water shrimp *Palaemon elegans* (*Rathke, 1837*) were collected from Saint-Malo Bay (France; 48˚64'N,−2˚00'W), in January 2018, using a shrimp hand net. Specimens were dissected and fixed as described above. Protocols for other species are described in the following: *Nebalia herbstii*, *Kenning et al. (2013)*; *Penaeus vannamei*, *Meth et al. (2017)*; *Saduria entomon*, *Kenning and Harzsch (2013)*; *Carcinus maenas*, *Krieger et al. (2012a)*; *Birgus latro*, *Krieger et al. (2010)*.

### Histology

The heads of Bouin-fixed animals (six specimens) were dehydrated in a graded series of ethanol and embedded in paraffin wax mixed with 5% beeswax. Serial sections (7 µm) were taken in the frontal or sagittal plane with a microtome (Leica RM 2145; Leica Microsystems, Wetzlar, Germany). The sections were stained with Azan-novum according to *Geidies (1954)* using standard protocols (*Welsch and Mulisch, 2010*).

### Immunohistochemistry

The brains of fixed animals (4% FA; five specimens) were dissected in PBS 0.1 M, pH 7.4, embedded in low-gelling agarose (Cat. A9414; Sigma-Aldrich Chemie GmbH, Munich, Germany) and sectioned (100 µm) with a vibratome (Hyrax V50; Carl Zeiss, Oberkochen, Germany). The sections were preincubated for 1.5 hr in PBT (PBS + 0.3% Triton X-100 +1% bovine serum albumine) to improve antibody penetration. Two sets of combinations of markers were used: 1. anti-synapsin +anti-allatostatin +nuclear marker; 2. anti-synapsin +anti-serotonin+nuclear marker. The sections were first incubated overnight in the primary antisera at room temperature. The antisera used were: monoclonal anti-SYNORF1 synapsin antibody (DSHB, 3C11; from mouse; 1:10 dilution; RRID: AB_2313867); polyclonal anti-A-allatostatin antiserum (A-type Dip-allatostatin I; Jena Bioscience, abd-062; from rabbit; 1:1000 dilution; RRID: AB_2314318); polyclonal anti-Serotonin (5-HT, Immunostar, Cat. No 20080, from rabbit, igG; 1:1000 dilution; RRID: AB_572263). After incubation, the sections were washed in several changes of PBT for 1 hr and afterwards incubated in the secondary antibodies (anti IgGs) conjugated to Alexa Fluor 488 (Alexa Fluor 488 goat anti-rabbit IgG Antibody, Invitrogen, Thermo Fisher Scientific; Waltham, MA, USA; RRID: AB_10374301) and Cy3 (Cy3-conjugated AffiniPure Goat Anti-Mouse IgG Antibody, Jackson ImmunoResearch Laboratories Inc.; West Grove, PA, USA; RRID: AB_2338000) overnight at room temperature. Additionally, HOECHST 33258 (Cat. 14530; Sigma-Aldrich Chemie GmbH, Munich, Germany) was used as a nuclear marker to show the cell clusters.

The sections were finally washed in several changes of PBT for 2 hr and mounted in Mowiol 4–88 (Cat. 0713.2; Carl Roth, Karlsruhe, Germany).

## Antibody specificity

### Synapsin

The monoclonal anti-SYNORF1 synapsin antibody (DSHB Hybridoma Product 3C11; anti SYNORF1 as deposited to the DSHB by E. Buchner, University Hospital Würzburg, Germany; supernatant) was raised against a *Drosophila melanogaster* GST-synapsin fusion protein and recognizes at least four synapsin isoforms (70, 74, 80 and 143 kDa) in western blots of *D. melanogaster* head homogenates (*Klagges et al., 1996*). *Sullivan et al. (2007)* mention a single band at approx. 75 kDa in a western blot analysis of crayfish brain homogenate. *Harzsch and Hansson (2008)* conducted a western blot analysis comparing brain tissue of *D. melanogaster* and the hermit crab *Coenobita clypeatus* (Anomura, Coenobitidae). The SYNORF1 serum provided identical results for both species and it stained one strong band between 80 and 90 kDa and a second weaker band slightly above 148 kDa, suggesting that the epitope that SYNORF1 recognizes is strongly conserved between *D. melanogaster* and *C. clypeatus* (see *Harzsch and Hansson, 2008*). Similar to the fruit fly, the antibody consistently labels brain structures in other major subgroups of the malacostracan crustaceans (e.g., *Beltz et al., 2003*; *Harzsch et al., 2002*; *Harzsch et al., 1999*; *Harzsch et al., 1998*; *Krieger et al., 2012b*) in a pattern that is consistent with the assumption that this antibody labels synaptic neuropils in crustaceans.

### Allatostatin

The A-type allatostatins represent a large family of neuropeptides that were first identified from the cockroach *Diploptera punctata*; they additionally share the C-terminal motif -YXFGLamide (*Christie et al., 2010*; *Nässel and Homberg, 2006*; *Stay et al., 1995*; *Stay and Tobe, 2007*). In the shore crab *Carcinus maenas* (Brachyura), almost 20 native A-type allatostatin-like peptides were identified from extracts of the thoracic ganglia (*Duve et al., 1997*). Shortly afterwards, various other A-type allatostatin-like peptides were isolated from the Eastern Crayfish *Orconectes limosus* (Astacida; *Dircksen et al., 1999*). Meanwhile, A-type allatostatin peptides have been discovered in a wide range of malacostracan crustaceans, including Brachyura (e.g. *Huybrechts et al., 2003*), Astacida (e.g. *Cape et al., 2008*), the prawns *Penaeus monodon* (*Duve et al., 2002*), *Macrobrachium rosenbergii* (*Yin et al., 2006*) and also in the shrimp *Penaeus vannamei* (*Ma et al., 2010*; *Meth et al., 2017*). *Christie (2016)* identified a total of 29 peptides with the C-terminal motif, -YXF-GLamide, in the latest analysis on the peptidome of the shore crab. The polyclonal rabbit allatostatin antiserum used in the present study was raised against the *Diploptera punctata* A-type Dip-allatostatin I,APSGAQRLYGFGLamide, coupled to bovine thyroglobulin using glutaraldehyde (*Vitzthum et al., 1996*). It has previously been used to localize A-type allatostatin-like peptides in crustacean and insect nervous systems (e.g., *Kreissl et al., 2010*; *Polanska et al., 2012*). In the following, the term 'allatostatin-like immunoreactivity' is used to indicate that the antibody most likely binds to various related peptides within this peptide family.

### Serotonin

The antiserum against serotonin (ImmunoStar Incorporated; Cat. No. 20080, Lot No. 541016) is a polyclonal rabbit antiserum raised against serotonin coupled to bovine serum albumin (BSA) with paraformaldehyde. The antiserum was quality control tested by the manufacturer using standard immunohistochemical methods. According to the manufacturer, staining with the antiserum was completely eliminated by pretreatment of the diluted antibody with 25 µg of serotonin coupled to BSA per ml of the diluted antibody. We repeated this control with the serotonin-BSA conjugate that was used for generation of the antiserum as provided by ImmunoStar (Cat. No. 20081, Lot No. 750256; 50 µg of lyophilized serotonin creatinine sulfate coupled to BSA with paraformaldehyde). Preadsorption of the antibody in working dilution with the serotonin-BSA conjugate at a final conjugate concentration of 10 µg/ml at 4°C for 24 hr completely blocked all immunolabeling. We performed an additional control and preadsorbed the diluted antiserum with 10 mg/ml BSA for 4 hr at room temperature. This preadsorption did not affect the staining, thus, providing evidence that the antiserum does not recognize the carrier molecule alone. The manufacturer also examined the cross

reactivity of the antiserum. According to the data sheet, with 5 µg, 10 µg, and 25 µg amounts, the following substances did not react with the antiserum diluted to 1:20,000 using the horse radish peroxidase (HRP) labeling method: 5-hydroxytryptophan, 5-hydroxyindole-3-acetic acid, and dopamine.

## Imaging

The brain tissues processed for immunofluorescence were viewed with a Leica TCS SP5II confocal laser-scanning microscope equipped with DPSS, Diode- and Argon-lasers and operated by the Leica 'Application Suite Advanced Fluorescence' software package (LASAF) (Leica Microsystems, Wetzlar, Germany). Digital images were processed with Adobe Photoshop CS4 or ImageJ. Only global picture enhancement features (brightness and contrast) were used.

The head tissues processed for histology were viewed with a Nikon Eclipse 90i upright microscope and bright-field optics (Nikon, Amstelveen, Netherlands). Serial images using a mounted digital camera (Nikon DS-Fi3) were aligned manually with the 3D-reconstruction software Amira 5.6.0 (FEI Visualization Science Group, Burlington, VT, USA; RRID: SCR_007353).

For frontal and sagittal sections, dorsal is always towards the top.

In the figures, the following color-coded abbreviations were used to identify the markers: SYN, synapsin (*magenta*); AstA, allatostatin (*green*); 5HT, serotonin (*green*); NUC, nuclear counter stain (*cyan*). Colors were chosen according to Color Universal Design for accessibility to colorblind readers.

## X-ray micro-computed tomography

Micro-CT scans were performed using an X-ray microscope (Xradia MicroXCT-200; Carl Zeiss Microscopy GmbH, Jena, Germany) that uses a 90-kV/8 W tungsten X-ray source and switchable scintillator-objective lens units as described by *Sombke et al. (2015)*. The heads of fixed animals (Bouin; two specimens) were contrasted in iodine solution (2% iodine resublimated (Cat. #X864.1; Carl Roth GmbH, Karlsruhe, Germany) in 99.5% ethanol), critical point-dried using a fully automatic critical point dryer Leica EM CPD300 (Leica Microsystems, Wetzlar, Germany) and scanned dry (scan medium air). Tomography projections were reconstructed using the reconstruction software XMReconstructor (Carl Zeiss Microscopy GmbH, Jena, Germany), resulting in image stacks (DICOM format) with a pixel size of about 5.8 µm for the 4 × objective and 1.9 µm for the 10 × objective.

## 3D reconstruction

The 3D reconstructions of brain and substructures are based on manual segmentation based on image stacks obtained either by the micro-CT scans or by the alignment of serial histological sections, and were performed using the software Amira (FEI Visualization Science Group, Burlington, VT, USA) as described in *Sombke et al. (2015)*. The computed 3D surfaces were slightly smoothed.

## Nomenclature

The neuroanatomical nomenclature used in this manuscript for neuropils, clusters of cell bodies and tracts is based on *Sandeman et al. (1993)* and *Richter et al. (2010)* with some modifications adopted from *Krieger et al. (2015)* and *Loesel et al. (2013)*. The term 'visual neuropils' is used instead of 'optic neuropils' as suggested by *Krieger et al. (2015)*. The terms lamina, medulla and lobula are used for the visual neuropils instead of the lamina ganglionaris, medulla externa and medulla interna (*Harzsch, 2002*). The term 'olfactory neuropil' refers to the deutocerebral chemosensory lobe in *Loesel et al. (2013)* and *Krieger et al. (2015)*. The olfactory globular tract is named the projection neuron tract (PNT) according to *Loesel et al. (2013)*. Cell clusters are referred by their given numbers in parentheses. Because no border was detectable between the cell clusters (9) and (11), they are collectively referred as cluster (9/11) (*Krieger et al., 2015*), and accordingly are the cell clusters (2) and (3), referred as cluster (2/3). (x) refers likely to the fusion of the cell clusters (12 , 13) and (17) according to the nomenclature from *Sandeman et al. (1992)*.

## Measurements

For the volumes of the HN-TM (hemiellipsoid body and terminal medulla complex in both hemispheres) and the olfactory neuropils relative to the total brain volume, measurements were made from 3D reconstructions of the relevant structures from micro-CT scans using the Amira software.

For the calculation of total brain volume of *P. elegans*, the volume of the tissues connecting the lateral protocerebrum with the central brain in the eye peduncles was omitted, because by omitting the neurites connecting both brain regions, the total brain volume is better comparable with that of *R. exoculata*. Same calculations were applied on the *Birgus latro* data from *Krieger et al. (2010)*.

The number of globuli cells (i.e. cell somata in the cell cluster (5)) was determined by estimation of the globuli cell densities in the cell cluster (5), and the total volume of one cell cluster (5). The globuli cell densities were estimated by direct counting of the somata within 0.02 to 0.04 mm$^2$ paraffin sections of 0.007 mm thickness ($1.3 \times 10^{-4}$ to $2.8 \times 10^{-4}$ mm$^3$), with a density estimated to be approximately $1.3 \times 10^6$ globui cells per mm$^3$. The total volume of one cell cluster (5) was calculated from 3D-reconstructions with the Amira software.

For the volume of the olfactory neuropils and the number of olfactory glomeruli, measurements and estimations were made from sections revealed by synapsin immunoreactivity as described in *Beltz et al. (2003)*.

## Data repository

The morphological raw data of our contribution have been deposited in the online repository MorphDBase (https://www.morphdbase.de/) under the following accession numbers :

J_Machon_20190502 M-3.1
J_Machon_20190502 M-4.1
J_Machon_20190502 M-5.1
J_Machon_20190502 M-6.1
J_Machon_20190502 M-7.1

## Acknowledgements

The authors thank the chief scientist of the Bicose 2018 cruise (M-A. Cambon), as well as the captain and crew of the research vessel 'Pourquoi pas?' and the HOV Nautile team for sampling the hydrothermal shrimp. We thank B Shillito and L Amand for the use of the PERISCOP device, the Ifremer for providing pictures of *R exoculata* swarms, E Becker for the histological sections, P O M Steinhoff for critical point drying the samples, M Charmantier for feedback on the organ of Bellonci, and C Wirkner and S Scholz for their advice on the myoarterial formation.

This study was financed by an incoming international stipend from the University of Greifswald, the German Science Foundation (Grant number: DFG INST 292/119–1 FUGG, DFG INST 292/120–1 FUGG), and a fellowship grant from InterRidge. We acknowledge support for the Article Processing Charge from the DFG (German Research Foundation, 393148499) and the Open Access Publication Fund of the University of Greifswald.

## Additional information

### Funding

| Funder | Grant reference number | Author |
| --- | --- | --- |
| Deutsche Forschungsgemeinschaft | DFG INST 292/119-1 FUGG | Steffen Harzsch |
| Deutsche Forschungsgemeinschaft | DFG INST 292/120-1 FUGG | Steffen Harzsch |
| Deutsche Forschungsgemeinschaft | 393148499 | Steffen Harzsch |
| InterRidge | Fellowship Program | Julia Machon |

The funders had no role in study design, data collection and interpretation, or the decision to submit the work for publication.

## Author contributions
Julia Machon, Conceptualization, Data curation, Formal analysis, Validation, Investigation, Visualization, Writing—original draft, Project administration, Writing—review and editing; Jakob Krieger, Formal analysis, Investigation, Visualization, Methodology, Writing—original draft, Writing—review and editing; Rebecca Meth, Data curation, Formal analysis, Writing—review and editing; Magali Zbinden, Conceptualization, Resources, Supervision, Funding acquisition, Investigation, Project administration, Writing—review and editing; Juliette Ravaux, Conceptualization, Data curation, Supervision, Funding acquisition, Methodology, Project administration, Writing—review and editing; Nicolas Montagné, Conceptualization, Data curation, Supervision, Investigation, Methodology, Writing—review and editing; Thomas Chertemps, Conceptualization, Resources, Data curation, Supervision, Investigation, Methodology, Writing—review and editing; Steffen Harzsch, Conceptualization, Resources, Formal analysis, Supervision, Writing—original draft, Writing—review and editing

## Author ORCIDs
Julia Machon https://orcid.org/0000-0003-3734-9313
Steffen Harzsch https://orcid.org/0000-0002-8645-3320

## Decision letter and Author response
Decision letter https://doi.org/10.7554/eLife.47550.029
Author response https://doi.org/10.7554/eLife.47550.030

# Additional files

## Supplementary files
• Transparent reporting form
DOI: https://doi.org/10.7554/eLife.47550.016

## Data availability
The raw data of the micro CT scans and the histological section series have been made public at the morphological data repository Morph D Base: https://www.morphdbase.de/. The data have been deposited under the accession numbers: J_Machon_20190502-M-3.1 J_Machon_20190502-M-4.1 J_Machon_20190502-M-5.1 J_Machon_20190502-M-6.1 J_Machon_20190502-M-7.1.

The following datasets were generated:

| Author(s) | Year | Dataset title | Dataset URL | Database and Identifier |
|---|---|---|---|---|
| Machon J, Krieger J, Meth R, Zbinden M, Ravaux J, Montagné N, Chertemps T, Harzsch S | 2019 | Data from: Neuroanatomy of a hydrothermal vent shrimp provides insights into the evolution of crustacean integrative brain centers | https://www.morphdbase.de/?J_Machon_20190502-M-3.1 | Morph D Base, J_Machon_20190502-M-3.1 |
| Machon J, Krieger J, Meth R, Zbinden M, Ravaux J, Montagné N, Chertemps T, Harzsch S | 2019 | Data from: Neuroanatomy of a hydrothermal vent shrimp provides insights into the evolution of crustacean integrative brain centers | https://www.morphdbase.de/?J_Machon_20190502-M-4.1 | Morph D Base, J_Machon_20190502-M-4.1 |
| Machon J, Krieger J, Meth R, Zbinden M, Ravaux J, Montagné N, Chertemps T, Harzsch S | 2019 | Data from: Neuroanatomy of a hydrothermal vent shrimp provides insights into the evolution of crustacean integrative brain centers | https://www.morphdbase.de/?J_Machon_20190502-M-5.1 | Morph D Base, J_Machon_20190502-M-5.1 |
| Machon J, Krieger J, Meth R, Zbinden M, Ravaux J, Montagné N, Chertemps T, Harzsch S | 2019 | Data from: Neuroanatomy of a hydrothermal vent shrimp provides insights into the evolution of crustacean integrative brain centers | https://www.morphdbase.de/?J_Machon_20190502-M-6.1 | Morph D Base, J_Machon_20190502-M-6.1 |
| Machon J, Krieger J, Meth R, Zbinden | 2019 | Data from: Neuroanatomy of a hydrothermal vent shrimp provides | https://www.morphdbase.de/?J_Machon_ | Morph D Base, J_Machon_20190502-M- |

| M, Ravaux J, Montagné N, Chertemps T, Harzsch S | insights into the evolution of crustacean integrative brain centers | 20190502-M-7.1 | 7.1 |

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

# Appendix 1

DOI: https://doi.org/10.7554/eLife.47550.017

## Abbreviations

(**Numbers 1-X**), cell clusters 1-X; **5HT**, serotonin immunoreactivity; **A1**, antenna 1; **A1l**, lateral flagellum of antenna 1; **A1lNv**, lateral antenna 1 nerve; **A1m**, medial flagellum of antenna 1; **A1mNv**, medial antenna 1 nerve; **A2**, antenna 2; **A2Nv**, antenna 2 nerve; **AcN**, accessory lobe/neuropil; **AMPN**, anterior medial protocerebral neuropil; **AnN**, antenna 2 neuropil; **ASTir**, allatostatin-like immunoreactivity; **b**, base region of the olfactory glomerulus; **bs**, branchiostegite; **c**, cap region of the olfactory glomerulus; **CA**, cerebral artery; **CA$_L$**, lateral cerebral artery; **CA$_M$**, median cerebral artery; **CB**, central body; **DC**, deutocerebrum; **dR**, degenerated rhabdoms; **ENv**, eye nerve; **HN**, hemiellipsoid body neuropil; **HN$_{cap}$**, hemiellipsoid body cap region; **HN$_{core}$**, hemiellipsoid body core region; **IL**, intermediate layer; **La**, lamina; **LAN**, lateral antenna 1 neuropil; **lF**, lateral foramen; **Lo**, lobula; **lPC**, lateral protocerebrum; **maf**, myoarterial formation; **maf$_m$**, myoarterial formation muscles; **MAN**, median antenna 1 neuropil; **MAR**, Mid-Atlantic Ridge; **Me**, medulla; **mF**, medial foramen; **mPC**, median protocerebrum; **MPN**, median protocerebral neuropil; **OA**, ophthalmic artery; **Ob**, onion bodies; **OBNv**, organ of Bellonci nerve; **oc**, oesophageal connectives; **ocp**, ocular plate; **og**, olfactory glomerulus; **ON**, olfactory neuropil; **PB**, protocerebral bridge; **pc**, cluster of pigment cells; **PMPN**, posterior medial protocerebral neuropil; **PNT**, projection neuron tract; **PNTN**, projection neuron tract neuropil; **PNTCN**, projection neuron tract central neuropil; **PT**, protocerebral tract; **R**, retina; **r**, rostrum; **sbc**, subcap region of the olfactory glomerulus; **sc**, scaphocerite; **SYNir**, synapsin immunoreactivity; **T**, tapetum; **T'$_d$**, dorsal secondary tendon; **T'$_v$**, ventral secondary tendon; **T$_a$**, anterior tendon; **T$_d$**, dorsal tendon; **TC**, tritocerebrum; **TM**, terminal medulla neuropil; **TN**, tegumentary neuropil; **TNv**, tegumentary nerve; **V**, vessel; **VN**, visual neuropils

