## [Decision Letter]

Thank you for submitting your article "Neuroanatomy of a hydrothermal vent shrimp provides insights into the evolution of crustacean integrative brain centers" for consideration by *eLife*. Your article has been reviewed by three peer reviewers, and the evaluation has been overseen by Eve Marder as the Senior and Reviewing Editor. The following individuals involved in review of your submission have agreed to reveal their identity: DeForest Mellon (Reviewer #1); Barb Beltz (Reviewer #3).

The reviewers have discussed the reviews with one another and the Reviewing Editor has drafted this decision to help you prepare a revised submission.

Usually at *eLife* we provide a consensus summary that describes the essential revisions. However, in your case, the three reviewers all agreed about the interest of the work, and most of what they are asking you to deal with requires minor textual and editorial modifications. Therefore, I am taking the unusual step of just including the three reviews in full, and asking you to follow their guidance to make your already interesting manuscript more accessible to the readership.

*Reviewer #1:*

This paper represents a thorough and highly detailed study of the brain structures of the hydrothermal vent shrimp, Rimacaris. The extreme sensory environment of this animal makes for a fascinating comparative study of CNS modifications; the bauplan of the Rimacaris brain is similar to that of other decapods, with some major exceptions. First and foremost, the optic peduncles are lost, and the structures of the lateral protocerebrum are in proximity to the median protocerebrum. (BTW, this modification is similar to the loss of optic peduncles and protocerebral condensation in another Caridean family, the Alpheids) The optic ganglia are reduced in volume, and the retinal areas of the compound eyes are elongated and fixed to the ventral surface of a transparent dorsal optic plate. The dioptric apparatus has been lost. Of significant importance is the hypertrophy of the hemiellipsoid bodies which, together with the terminal medulla, occupy roughly 25% of the entire brain volume. The authors hypothesize, supported by several other studies, that this is an adaptation for enhanced spatial memory, a necessity due to the reduction in the visual and olfactory systems in these animals.

This study is profusely illustrated with immunochemical and x-ray micro-computed tomography-generated 3-D images, with are extremely helpful in understanding the authors' observations and discussion points. The immunohistologicaal studies for synapsin, allostatin and serotonin are well controlled.

I have no critical comments about this study; it is well conceived and executed. The discussion is pertinent. Furthermore, the senior author, Steffan Harzsch, is a world renowned expert on crustacean neuroanatomy and the evolution of the crustacean nervous systems. I highly recommend its publication in *eLife*.

*Reviewer #2:*

This manuscript describes the anatomy of the brain of the hydrothermal vent shrimp Rimicaris exoculata. This shrimp has been of special interest to scientists since the discovery of it and other specialized organisms in deep sea hydrothermal vents some 35 years ago. Given the specializations of this shrimp and other species to this extraordinary environment, these species have been elevated from their otherwise rather "modest invertebrate" status to the category of "charismatic megafauna"…a plus for a journal that publishes this work. The research team combines scientists investigating various aspects of these shrimp with other scientists who are specialists in neuroanatomy, and the collaboration pays off – they provide a very thorough and high quality description of the neuroanatomical organization of the brain of this shrimp.

A couple of comments:

1) The comparisons of relative sizes of the olfactory neuropils and "integrative centers" (hemiellipsoid body and terminal medulla) could be made more explicit, in order to support the authors' points. For example, in subsection “The olfactory system”, the authors state that "The olfactory neuropils of R. exoculata are relatively small in terms of volume and not overly developed compared to other species (Table 1…. "). They also make this point in subsection “Possible functions of the hemiellipsoid body: new lesson from R. exoculata?”. From the data presented in Table, this statement does not seem to be so evident. Yes, the olfactory neuropil of Rimicaris has a volume (56) that is half the volume (120) of Palaemon, another caridean shrimp of about the same body size and of the stomatopod (110). But they do not point out that according to this table, the olfactory neuropil volume for Rimicaris is larger than that of some other crustaceans that are of even larger body size than Rimicaris, such as Procambarus (10) and Saduria (3). The comparison between Rimicaris and other species on the list is more difficult because of much larger differences in body size. So, according to this analysis, the data as presented do not make such a compelling case in support of this statement that the olfactory neuropils of Rimicaris are relatively small. Likewise, the statement is made that the hemiellipsoid bodies and terminal medulla of Rimicaris have volumes that are relative large compared to other crustaceans. The authors offer the images on Figure 11 as supporting this, but from these single images it is difficult to quickly and quantitatively see this. Beyond Figure 11, they only offer three numbers in the text (subsection “Comparisons of the olfactory system and the higher integrative centers with other crustaceans”): that these centers occupy 25% of total brain volume in Rimicaris, 22% in Palaemon, and 13% in Birgus. So, the size of these centers in Rimicaris is no different than in the other caridean shrimp (Palaemon) that lives in a different environment. It is not clear then why the Rimicaris integrative centers are "remarkable" if it is the same as an unremarkable caridean shrimp. In addition, why just compare the size with one other non-caridean crustacean (Birgus)? What about providing a comparison with other species, as was done for the olfactory neuropil in Table 1, to make the case that the difference in relative size is "remarkable"?

2) There should be some concern about concluding too much about functional differences from any differences in size of structures. There are many cautionary tales on this. Striedter's 2005 book, Principles of Brain Evolution, provides many cautionary examples. Along these lines, stating in the Abstract that because the "remarkable" (Discussion first paragraph) hemiellipsoid bodies are relatively large, they "must fulfil functions in addition to higher order sensory processing" (emphasis on "must") (also in subsection “Possible functions of the hemiellipsoid body: new lesson from R. exoculata?”) is a reach. It is fine to speculate, but "must" seems to be an attempt to oversell it. And the speculation on the involvement of these integrative centers in place memory is fine. But it is just speculation. Furthermore, it is not obvious that this is even a testable or falsifiable hypothesis…certainly not easily doable.

*Reviewer #3:*

This article identifies and describes regions in the brain of the alvinocaridid shrimp Rimicaris exoculate. The morphological features of these regions are then interpreted in the context of (1) an evolutionary framework within malacostracan crustaceans, and (2) inferred functional abilities and constraints related to their unusual habitat in deep hydrothermal vents. I find this work exciting and significant due to the intrinsic interest in understanding how organisms adapt successfully to extreme environments such as those found in hydrothermal vents, as well as contributions to our understanding of malacostracan brain evolution and potential functions of specific brain regions whose roles are currently not fully understood.

The greatest strengths of this manuscript are:

1) The careful descriptions of brain regions and associated documentation in elegant images of the brain using three technical approaches: histology, immunocytochemistry and X-ray micro-computer tomography. The figures strongly support and in some areas amplify the results, allowing the reader to get an observer's appreciation for the spatial relationships and microanatomy of specific brain regions. The addition of CT scans to the more conventional histological and antibody-based methods provides a topological overview that is then woven through the narrative (and figures), providing an overview that is often lacking in these types of articles, except through the use of drawings that seldom have the accuracy or quantitative information found in direct CT scans. The text is for the most part readily accessible to the reader because the authors provide a fairly simple roadmap of the brain based largely on the CT scans (e.g., see Figure 3 and color-coded key of brain regions).

2) The extensive review of previous literature about the brains of related species, which provides a broad context for the new R. exoculata data. The comparisons that are drawn allow the authors to conclude that olfactory abilities have not expanded to compensate for the lack of visual acuity in this species, as one might expect. Rather, the enlargement of regions in the lateral protocerebrum ---in particular the organ of Bellonci and hemiellipsoid bodies--- appear to be among the primary brain adaptations for success in the hydrothermal habitat; the cor frontale also is enlarged and is likely to be able to pump blood more efficiently. While most of the specific functional ramifications of the anatomical observations cannot be known, this manuscript nevertheless lays the careful groundwork for further studies by identifying questions that persist, such as the role of the organ of Bellonci in sensory reception and the reasons why the hemiellipsoid body undergoes rather extreme anatomical modifications during evolution (see Figure 12). Quantitative features of the primary olfactory processing regions of malacostracan species also are compared and substantiate the conclusion that major changes in these centers in R. exoculata do not offset the loss of visual abilities (Table 1). The suggestion that the limited visual abilities may be utilized for detecting thermal radiation is not new, but nevertheless a fascinating possibility.

While I found no major weaknesses in this manuscript, there are areas that can be improved.

Although the authors have done an admirable job of explaining the brain anatomy of R. exoculata, there are some spots in the text where I believe some re-wording is necessary to convey the intended meaning clearly (e.g., see #1 below).

While the text makes extensive references to the figures, there are nevertheless occasional places where additional labels are needed on the figures so that the reader can identify features that are highlighted in the text. See examples below.

Specific comments:

1) Subsection “Comparisons of the olfactory system and the higher integrative centers with other crustaceans”. The thoughts here become confused and I believe the second part of the sentence technically misstates what the authors mean to say, although I suppose I suggest first of all splitting this very long sentence into two, ending with "…the giant robber crab Birgus latro (13%)." I believe the second part of this sentence intends to say "However, the olfactory neuropils of P. elegans and B. latro occupy roughly two and six times more volume than these regions R. exoculata, values that represent 4.2%, 17% and 2.7% of total brain volume respectively in these species." I think it's worth spelling out these relationships, even though it takes a few extra words. These are important points for the conclusions.

2) Subsection “The myoarterial formation and cerebral vascular system”: Presumably the authors also observed elements of the hematopoietic system, as these are closely associated with the cor frontale. Can the locations of hematopoietic tissues be identified in this paper (this would be very helpful information even if only a brief statement)?

3) Subsection “A visual system adapted to a dim light environment” paragraph two: Can this anatomy actually imply retinotopy without dye fills or physiological studies to demonstrate this relationship?

4) Subsection “A visual system adapted to a dim light environment” paragraph three: Is a goal of selective pressures the limitation in the amount of energy used? Or simply eliminating structures that are no longer useful (which may secondarily accomplish the metabolic conservation you suggest)?

5) Subsection “Evolution of higher integrative brain centers: the hemiellipsoid body” paragraph two: What is the meaning of "secondary sensory input"?

6) Section on immunocytochemistry: Also provide the antibody dilutions that were used.

---

## [Author Response]

Usually at eLife we provide a consensus summary that describes the essential revisions. However, in your case, the three reviewers all agreed about the interest of the work, and most of what they are asking you to deal with requires minor textual and editorial modifications. Therefore, I am taking the unusual step of just including the three reviews in full, and asking you to follow their guidance to make your already interesting manuscript more accessible to the readership.

Reviewer #2:

[…] A couple of comments:1) The comparisons of relative sizes of the olfactory neuropils and "integrative centers" (hemiellipsoid body and terminal medulla) could be made more explicit, in order to support the authors' points. For example, in subsection “The olfactory system”, the authors state that "The olfactory neuropils of R. exoculata are relatively small in terms of volume and not overly developed compared to other species (Table 1…. "). They also make this point in subsection “Possible functions of the hemiellipsoid body: new lesson from R. exoculata?”. From the data presented in Table, this statement does not seem to be so evident. Yes, the olfactory neuropil of Rimicaris has a volume (56) that is half the volume (120) of Palaemon, another caridean shrimp of about the same body size and of the stomatopod (110). But they do not point out that according to this table, the olfactory neuropil volume for Rimicaris is larger than that of some other crustaceans that are of even larger body size than Rimicaris, such as Procambarus (10) and Saduria (3). The comparison between Rimicaris and other species on the list is more difficult because of much larger differences in body size. So, according to this analysis, the data as presented do not make such a compelling case in support of this statement that the olfactory neuropils of Rimicaris are relatively small.

Although the olfactory neuropils of *Rimicaris* are not the smallest ones, the important piece of information here is that they are not overly developed compared to other species. We may expect that they are overly developed to compensate for poor visual ability but this is not the case. However, following the reviewer’s suggestion to improve this statement, we modified the Results section accordingly (subsection “Comparisons of the olfactory system and the higher integrative centers with other crustaceans”) including a more detailed comparisons from Table 1, and in the Discussion. We toned down our statement by changing “small” to “moderately developed”. Nonetheless, for the following on the volumes of the higher integrative centers, the statement that the olfactory neuropils of *Rimicaris* are relatively smalls is of major importance for the comparison to its close caridean relative *Palaemon*.

As pointed out, comparisons with some of the species listed in Table 1 are not really meaningful and may dilute part of the information. However, we believe such semi-exhaustive updated list nevertheless is useful for readers from the field, and also is consistent with the species presented in our phylogram (Figure 12).

Likewise, the statement is made that the hemiellipsoid bodies and terminal medulla of Rimicaris have volumes that are relative large compared to other crustaceans. The authors offer the images on Figure 11 as supporting this, but from these single images it is difficult to quickly and quantitatively see this. Beyond Figure 11, they only offer three numbers in the text (subsection “Comparisons of the olfactory system and the higher integrative centers with other crustaceans”): that these centers occupy 25% of total brain volume in Rimicaris, 22% in Palaemon, and 13% in Birgus. So, the size of these centers in Rimicaris is no different than in the other caridean shrimp (Palaemon) that lives in a different environment. It is not clear then why the Rimicaris integrative centers are "remarkable" if it is the same as an unremarkable caridean shrimp.

Figure 11 rather shows that, compared to other crustaceans, the integrative centers of *Rimicaris* are well-developed relative to the size of the olfactory neuropils, as stated in subsection “Comparisons of the olfactory system and the higher integrative centers with other crustaceans”. Because of the brain architecture in this species, horizontal sections (as we did for all the immunostainings on *Rimicaris*) cannot show conclusively the entire integrative centers. Therefore, we selected a picture showing only a small part of the hemiellipsoid body but also the terminal medulla to be consistent with the pictures presented for other species.

In terms of relative volume, and compared to *Palaemon*, we consider the integrative centers of *Rimicaris* “remarkable” because, in spite of smaller olfactory and visual inputs, they occupy roughly the same, even slightly higher volume of the brain than in *Palaemon*. This is an interesting feature since the volume of the integrative centers has been suggested to mirror the amount of sensory, especially olfactory, input. However, we are fully aware of the fact that, in the absence of more replicates, our study can only point out trends and cannot replace a sound volumetric analysis. Also, regarding the hemiellipsoid bodies, they can be considered “remarkable” too only from a neuroanatomical view: they are conspicuous, well-defined with a clear subdivision in cap/core regions, they contain large fiber tracts and are associated to a massive cell cluster…

Nevertheless, following this reviewer’s comments, we toned down our statement by deleting “remarkable” throughout the text and we clarified the conclusion.

In addition, why just compare the size with one other non-caridean crustacean (Birgus)? What about providing a comparison with other species, as was done for the olfactory neuropil in Table 1, to make the case that the difference in relative size is "remarkable"?

We obtained the volumetric information for the higher integrative neuropils from 3D reconstructions from microCT scans, which we performed on *Rimicaris* and *Palaemon*, and we had access to previous microCT data for *Birgus* only. Furthermore, these three species were processed with the same protocol. To use microCT data for species for which different fixation methods have been used would be inaccurate, because it can severely influence the results (e.g. 5% [ZnFa] to 12.5% [2%PFA/2%Glut] less volume in siblings of Marmorkrebs’ nervous tissus; Nischik and Krieger, 2018).

Nischik and Krieger, 2018, Evaluation of standard imaging techniques and volumetric preservation of nervous tissue in genetically identical offspring of the crayfish Procambarus fallax cf. virginalis (Marmorkrebs). *PeerJ*, *6*, e5181.

2) There should be some concern about concluding too much about functional differences from any differences in size of structures. There are many cautionary tales on this. Striedter's 2005 book, Principles of Brain Evolution, provides many cautionary examples. Along these lines, stating in the Abstract that because the "remarkable" (Discussion first paragraph) hemiellipsoid bodies are relatively large, they "must fulfil functions in addition to higher order sensory processing" (emphasis on "must") (also in subsection “Possible functions of the hemiellipsoid body: new lesson from R. exoculata?”) is a reach. It is fine to speculate, but "must" seems to be an attempt to oversell it. And the speculation on the involvement of these integrative centers in place memory is fine. But it is just speculation. Furthermore, it is not obvious that this is even a testable or falsifiable hypothesis…certainly not easily doable.

In the Abstract, we toned down our statement by replacing “must” by “may”, as the same for the Discussion subsection “Possible functions of the hemiellipsoid body: new lesson from R. exoculata?”. We fully agree that functional interpretations from differences in structures can be hazardous, and accordingly we tried to be as cautious as possible. Nevertheless the case of the olfactory neuropils in blind cave crustaceans is a nice example of how neuropil size differences can actually lead to functional interpretation. Considering the possible role of the integrative centers in place memory, it is indeed not worth considering to test this hypothesis for now because of technical and biological constraints, but hopefully future functional studies on more accessible models could one day provide new highlights to answer this question.

We added a sentence in the Discussion (subsection “Possible functions of the hemiellipsoid body: new lesson from R. exoculata?”).

Reviewer #3:

Specific comments:1) Subsection “Comparisons of the olfactory system and the higher integrative centers with other crustaceans”. The thoughts here become confused and I believe the second part of the sentence technically misstates what the authors mean to say, although I suppose I suggest first of all splitting this very long sentence into two, ending with "…the giant robber crab Birgus latro (13%)." I believe the second part of this sentence intends to say "However, the olfactory neuropils of P. elegans and B. latro occupy roughly two and six times more volume than these regions R. exoculata, values that represent 4.2%, 17% and 2.7% of total brain volume respectively in these species." I think it's worth spelling out these relationships, even though it takes a few extra words. These are important points for the conclusions.

Modified as proposed in subsection “Comparisons of the olfactory system and the higher integrative centers with other crustaceans”.

2) Subsection “The myoarterial formation and cerebral vascular system”: Presumably the authors also observed elements of the hematopoietic system, as these are closely associated with the cor frontale. Can the locations of hematopoietic tissues be identified in this paper (this would be very helpful information even if only a brief statement)?

We reviewed the histological section series of the studied specimens (as are now publicly available in the online repository MorphDBase https://www.morphdbase.de/ under accession numbers J_Machon_20190502-M-3.1

Unfortunately, we were unable to identify possible hematopoietic tissues with any certainty from the available material. I am afraid that we may need additional experiments as e.g., Shown in Da Silva et al., 2013 to answer this reviewer’s question but unfortunately are running out of specimens. However, we will pay attention to this issue in an upcoming manuscript on other vent shrimp specie's brains for which we still have more specimens at hand. Indeed, the question on an evolutionary origin of the HPT in the malacostracan lineage is highly interesting.

3) Subsection “A visual system adapted to a dim light environment” paragraph two: Can this anatomy actually imply retinotopy without dye fills or physiological studies to demonstrate this relationship?

The dorsal expansion of both the flattened retina and lamina suggests a retinotopy but isn’t a proof of it; accordingly we only formulate a hypothesis. We have modified the sentence. The issue to test this is that living specimens of *Rimicaris* are available only shipboard just after sampling, and are stored in hyperbaric aquaria for functional studies. Specimens at atmospheric pressure are in bad state and survive only few hours at most. Backfills from the retina may be feasible on freshly sampled animals, but electrophysiological experiments are very challenging (one successful electroretinogram performed by Johnson et al., 1995).

4) Subsection “A visual system adapted to a dim light environment” paragraph three: Is a goal of selective pressures the limitation in the amount of energy used? Or simply eliminating structures that are no longer useful (which may secondarily accomplish the metabolic conservation you suggest)?

We have modified this sentence as suggested by the reviewer.

5) Subsection “Evolution of higher integrative brain centers: the hemiellipsoid body” paragraph two: What is the meaning of "secondary sensory input"?

We have modified this sentence.

6) Section on immunocytochemistry: Also provide the antibody dilutions that were used.

Done in subsection “Histology”.